# MOMENT DISTRIBUTIONALLY ROBUST PROBABILISTIC SUPERVISED LEARNING

## ABSTRACT

Probabilistic supervised learning assumes the groundtruth itself is a distribution instead of a single label, as in classic settings. Common approaches learn with a proper composite loss and obtain probability estimates via an invertible link function. Typical links such as the softmax yield restrictive and problematic uncertainty certificates. In this paper, we propose to make direct prediction of conditional label distributions from first principles in distributionally robust optimization based on an ambiguity set defined by feature moment divergence. We derive its generalization bounds under mild assumptions. We illustrate how to manipulate penalties for underestimation and overestimation. Our method can be easily incorporated into neural networks for end-to-end representation learning. Experimental results on datasets with probabilistic labels illustrate the flexibility, effectiveness, and efficiency of this learning paradigm.

## 1 INTRODUCTION

The goal of classical supervised learning is point estimation—predicting a single target from the label domain given features—usually without justifying the confidence. The outcome distribution of an event can be inherently uncertain and more desirable than point predictions in some scenarios. For example, weather predictions that express the uncertainty of events such as rain occurring are more sensible than binary-valued predictions, while a uniform distribution prediction for the outcome of a fair dice roll is more sensible than speculating an integral value randomly. On one hand, the predicted distribution quantifies label uncertainty and is thus more informative than a point prediction, which is widely studied in weakly supervised learning (Yoshida et al., 2021), boosting (Friedman et al., 2000) and optimal treatment (Leibovici et al., 2000). On the other hand, the ground truth naturally comes with multiple targets, possibly with different importances. For instance, there can be multiple emotions in a human face image, there are different gene expression levels over a period of time in biological experiments, and many annotators might disagree over a highly ambiguous instance. In the above settings, each predefined label is part of the ground truth as long as it has a positive probability in the true distribution. Hence, it is natural to use probabilistic labels in both training and inference when the ground truth is no longer a point. In the literature, the task of predicting full distributions from features is called probabilistic supervised learning (Gressmann et al., 2018).

A probabilistic supervised learning task comes with a probabilistic loss functional quantitatively measuring the utility of the prediction (Bickel, 2007). Williamson et al. (2016) propose a composite multiclass loss that separates properness and convexity. They illuminate the connection between classification calibration (Tewari & Bartlett, 2007) and properness (Gneiting & Raftery, 2007; Dawid, 2007), representing Fisher consistency for classification and probability estimation respectively. A proper loss is minimized when predictions match the true underlying probability, which implies classification calibration, but not vice versa. Among proper losses, the logarithmic loss (Good, 1952) severely penalizes underestimation of rare outcomes and assessing the "surprise" of the predictor in an information-theoretic sense, the Brier score—originally proposed for evaluating weather forecasts (Brier, 1950)—is useful for assessing prediction calibration, and the spherical scoring rule (Bickel, 2007) is used when a distribution with lower entropy is desired. A single proper loss is sometimes not sufficient for scenarios that elicit optimistic or pessimistic predictions for decision making with practical concerns (Elsberry, 2002; Chapman, 2012). For example, underestimating disastrous events may provide very low utility, motivating more pessimistic predictions.

Therefore it is desirable for a proper loss to be flexible in its penalties for deviated predictions that combine statistical properties of multiple losses.

Deep neural networks typically adopt the softmax function to predict a legal distribution. However, softmax intentionally renormalizes the logits and therefore assumes that it follows a logistic distribution (Bendale & Boult, 2016). It is poor at calibration, uncertainty quantification and robustness against overfitting (Joo et al., 2020). The inverse of the canonical link function in Williamson et al. (2016) can be used to recover probabilities but commonly resembles softmax (Zou et al., 2008).

In this paper, we propose a probabilistic supervised learning method from first principles in distributionally robust optimization (DRO) for general proper losses that realize desired prediction properties. Instead of specifying a parametric distribution, it starts with a minimax learning problem in which the predictor non-parametrically minimizes the the most adverse risk among all distributions in an ambiguity set defined by empirical feature moments. The ambiguity set represents our uncertainty about the underlying distribution. By strong duality, we show that the primal DRO problem is equivalent to a regularized empirical risk minimization (ERM) problem. The regularization results naturally from the ambiguity set instead of being explicitly imposed. The ERM form also allows us to derive generalization bounds and make inferences from unseen data. We illustrate a set of solutions for general proper losses satisfying certain mild conditions and an efficient algorithm for a weighted sum of two common strictly proper losses. We conduct experiments on real-world datasets by adapting our method to end-to-end differentiable learning. We defer all technical proofs to the appendix.

**Contributions.** Our contributions are summarized as follows. (1) We propose a distributionally robust probabilistic supervised learning method. (2) We characterize the solutions to the proposed method and present an efficient algorithm for specific losses. (3) We incorporate our method into neural networks and perform extensive empirical study on real-world data.

## 1.1 RELATED WORK

Model assessment of probabilistic models via predictive likelihood has been studied in Bayesian models (Gelman et al., 2014), probabilistic forecasting (Gneiting & Raftery, 2007), machine learning (Masnadi-Shirazi & Vasconcelos, 2009), conditional density estimation (Sugiyama et al., 2010), information theory (Reid & Williamson, 2011) and representation learning (Dubois et al., 2020). A comprehensive framework for probabilistic supervised learning can be found in Gressmann et al. (2018).

Techniques developed to explicitly tackle multiclass probabilistic classification include multiclass logistic regression (Collins et al., 2002), support vector machines (Lyu et al., 2019; Wang et al., 2019), learning from noisy labels (Zhang et al., 2021), weakly supervised learning (Yoshida et al., 2021), and neural networks (Papadopoulos, 2013; Gast & Roth, 2018). Multilabel classification, aimed at predicting multiple classes with equal importance, has been analyzed by Cheng et al. (2010) and Geng (2016) in a general probabilistic setting. Note that confidence calibration (Guo et al., 2017) has a different objective from probabilistic supervised learning.

Fisher consistency results have been established for classification losses (Tewari & Bartlett, 2007), structured losses (Ciliberto et al., 2016; Nowak et al., 2020), proper losses (Williamson et al., 2016) and Fenchel-Young losses (Blondel et al., 2020).

The emerging field of DRO has led to learning methods with ambiguity sets defined by feature moments (Farnia & Tse, 2016; Mazuelas et al., 2020), $\phi$-divergence (Duchi & Namkoong, 2019) and the Wasserstein distance (Shafieezadeh-Abadeh et al., 2019). The moment-based ambiguity set adopted in this work originates from maximum entropy (Cortes et al., 2015; Mazuelas et al., 2022), with similar work studying classification (Asif et al., 2015; Fathony et al., 2016) and structured prediction (Fathony et al., 2018a;b).

## 2 PRELIMINARIES

### 2.1 NOTATIONS

We adopt the following notations by convention. A bold letter $\boldsymbol{x}$ denotes a vector whereas a normal letter $x$ represents a scalar. $x_i$ or $(\boldsymbol{x})_i$ stands for the $i$-th coordinate of $\boldsymbol{x}$. We denote random variables with capitalization (e.g, $X$ or $\boldsymbol{X}$) and sets with calligraphic capitalization (e.g., $\mathcal{X}$, $\mathcal{A}$). We denote by $[n]$ the set $\{1, 2, \ldots, n\}$. $|\cdot|$ means the absolute value of a scalar or the cardinality of a set, depending on the context. The $\ell_p$ norm of a vector is defined as $\|\boldsymbol{x}\|_p \triangleq (\sum_i |x_i|^p)^{1/p}$. The indicator function of a subset $\mathcal{S}$ of a set $\mathcal{X}$ is a mapping $\mathbb{I}_{\mathcal{S}} : \mathcal{X} \to \{0, 1\}$ such that $\mathbb{I}_{\mathcal{S}}(x) = 1$ if $x \in \mathcal{S}$ and $\mathbb{I}_{\mathcal{S}}(x) = 0$ otherwise. $\mathbb{I}(\cdot)$ is adopted for events so that $\mathbb{I}(\mathcal{S}) = 1$ if event $\mathcal{S}$ occurs and $\mathbb{I}(\mathcal{S}) = 0$ otherwise. We write $\delta_z$ as the Dirac point measure at $z \in \mathcal{Z}$. A probability simplex of $(d+1)$-dimensional vectors is represented as $\Delta^d$, whose superscript is omitted when the context is clear. We denote by $\mathcal{P}(\mathcal{Z})$ the set of all probability distributions on a set $\mathcal{Z}$.

### 2.2 PROBABILISTIC LOSS FUNCTIONALS

A loss function measures the quality of a prediction associated with an event. Scoring rules are widely adopted to assess probabilistic predictions, but can be naturally translated to loss functions by appropriate negation and normalization. To illustrate some examples, we consider a decision problem in which $y \in \mathcal{Y}$ is an outcome and $\mathbb{P}_Y \in \mathcal{P}(\mathcal{Y})$ is a predicted distribution over $\mathcal{Y}$. We denote by $\mathbf{p}_Y \triangleq (\mathbb{P}_Y(y))_{y \in \mathcal{Y}}^T$ a vector of probabilities.

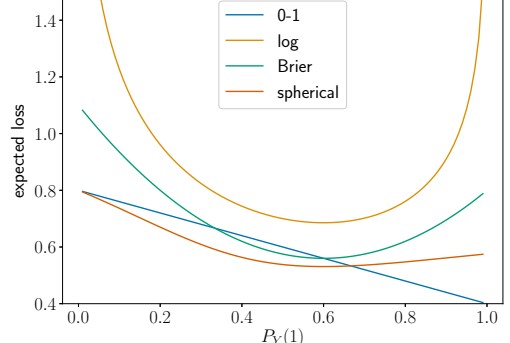

The **zero-one loss** is defined for deterministic prediction so that a penalty of 1 is incurred whenever $y'$ and $y$ differ: $\ell_{01}(y', y) \triangleq \mathbb{I}(y' \neq y)$. It extends to probabilistic predictions as $\ell_{01}(\mathbb{P}_Y, y) \triangleq 1 - \mathbb{P}_Y(y)^1$. The **cost-sensitive loss** for multiclass classification is similarly defined with a confusion cost matrix $\mathbf{C} \in \mathbb{R}_+^{|\mathcal{Y}| \times |\mathcal{Y}|}$: $\ell_{\mathrm{cs}}(\mathbb{P}_Y, y) \triangleq \sum_{i \in \mathcal{Y}} \mathbb{P}_Y(i) C_{iy}$.

Figure 1: The expected value of four loss functions for three classes with $\mathbb{Q}_Y(1) = 0.6$ and $\mathbb{Q}_Y(2) = \mathbb{Q}_Y(3) = 0.2$. $\mathbb{P}_Y(2) = \mathbb{P}_Y(3)$ as $\mathbb{P}_Y(1)$ varies. Each loss is normalized to cross $(1, 0)$ and $(0.5, 0.5)$ according to the binary case with a hard label. Best viewed in color.

The multiclass **Brier loss**, based on the Brier score or quadratic scoring rule, measures the mean squared difference between $\mathbb{P}_Y$ and $y$: $\ell_{\mathrm{br}}(\mathbb{P}_Y, y) \triangleq \sum_{y'} (\mathbb{P}_Y(y') - \mathbb{I}(y' = y))^2$.

The **logarithmic loss**, also called log-likelihood loss, incurs a rapidly increasing penalty as the predicted probability of the target event approaches zero: $L_{\log}(\mathbb{P}_Y, y) \triangleq -\ln \mathbb{P}_Y(y)$.

The **spherical scoring rule** can be interpreted as the spherical projection of the true belief onto the prediction vector. To use it as a loss function, we define $\ell_{\mathrm{sp}}(\mathbb{P}_Y, y) \triangleq 1 - \mathbb{P}_Y(y)/\|\mathbf{p}_Y\|_2$.

For ease of exposition, we define $L(\mathbb{P}, \mathbb{Q}) := \sum_y \mathbb{Q}_Y(y)\ell(\mathbb{P}_Y, y)$ where $\ell(\cdot, \cdot) : \mathcal{P}(\mathcal{Y}) \times \mathcal{Y} \to \mathbb{R}_+$ is a probabilistic loss function as illustrated above. A loss $L$ is called **proper** if $L(\mathbb{Q}, \mathbb{Q}) \leq L(\mathbb{P}, \mathbb{Q})$ for all $\mathbb{P}, \mathbb{Q}$, and called **strictly proper** if $\mathbb{Q}$ is the unique minimizer of $L(\cdot, \mathbb{Q})$. Figure 1 provides a graphical comparison of the above losses for prediction with three classes. We can infer that the zero-one loss is an improper loss.

### 2.3 PROBABILISTIC SUPERVISED LEARNING

We study the probabilistic supervised learning task where we are given $n$ training samples $\{(\mathbf{x}^{(1)}, y^{(1)}), (\mathbf{x}^{(2)}, y^{(2)}), \ldots, (\mathbf{x}^{(n)}, y^{(n)})\}$ drawn i.i.d. from a distribution $\mathbb{P}$ on the joint space

---

[1]In the literature, the zero-one loss is sometimes defined as $\ell_{01}(\mathbb{P}_Y, y) := \mathbb{I}(y \notin \arg\max_{y'} \mathbb{P}_Y(y'))$, which is proper, but discontinuous and not strictly proper.

$\mathcal{X} \times \mathcal{Y}$, in which $\mathcal{X}$ is a feature space and $\mathcal{Y}$ is a univariate finite discrete label space. A probabilistic multiclass loss function $L : \mathcal{P}(\mathcal{Y}) \times \mathcal{P}(\mathcal{Y}) \to \mathbb{R}_+$ is given. The goal of ERM is to learn from the samples a mapping $h : \mathcal{X} \to \mathcal{P}(\mathcal{Y})$ to minimize the empirical $L$-risk of $h$:

$$h^* \in \arg\min_{h \in \mathcal{H}} R_{\mathbb{P}^{\text{emp}}}^L(h) := \mathbb{E}_{\mathbb{P}_{\mathbf{X}}^{\text{emp}}} \left[ L(h(\mathbf{X}), \mathbb{P}_{Y|\mathbf{X}}^{\text{emp}}) \right], \tag{1}$$

where $\mathbb{P}_{\mathbf{X},Y}^{\text{emp}}$ represents the empirical distribution and $\mathcal{H}$ is a hypothesis space. Here we assume $\mathbf{x}$ may be accompanied with a probabilistic label by aggregating instances with the same $\boldsymbol{x}^{(i)}$. In this way, both learning and inference are accomplished in the general setting subsuming classical supervised learning.

## 3 METHOD

We now present our formulation for learning with general multiclass probabilistic losses. We provide theoretical results of consistency and generalization. We study the solution for general proper losses in our formulation and develop an efficient algorithm for two typical proper losses.

### 3.1 FORMULATION

We consider a continuous proper loss $L$ to be optimized under the unknown distribution $\mathbb{P}^{\text{true}}$. We assume that a class-sensitive feature function $\boldsymbol{\phi} : \mathcal{X} \times \mathcal{Y} \to \mathbb{R}^d$ that maps a data point to a $d$-dimensional feature vector is given. Examples include the multi-vector representation and class-dependent TF-IDF scores. Choosing a good $\boldsymbol{\phi}$ is a representation learning problem, but as we will discuss in Section 3.4, it is not a concern once our method is incorporated into neural networks as a layer. Intuitively, the elements of the vector $\boldsymbol{\phi}(\mathbf{x}, y)$ can be regarded as scores indicating how well the label $y$ matches with the feature $\mathbf{x}$. For example, with a linear hypothesis $h_{\mathbf{w}}(\mathbf{x}, y) = \langle \mathbf{w}, \boldsymbol{\phi}(\mathbf{x}, y) \rangle$, a good parameter vector $\mathbf{w}^*$ should yield

$$\langle \mathbf{w}^*, \boldsymbol{\phi}(\mathbf{x}, y) \rangle > \langle \mathbf{w}^*, \boldsymbol{\phi}(\mathbf{x}, y') \rangle \implies \mathbb{P}(\mathbf{x}, y) > \mathbb{P}(\mathbf{x}, y').$$

Instead of specifying a parametric form of predictions, we adopt a minimax statistical learning formulation:

$$\min_{\mathbb{P}_{Y|\mathbf{X}} \in \mathcal{P}(\mathcal{Y})} \max_{\mathbb{Q} \in \mathcal{A}(\mathbb{P}^{\text{emp}})} \mathbb{E}_{\mathbb{Q}_{\mathbf{X}}} \left[ L\left( \mathbb{P}_{Y|\mathbf{X}}, \mathbb{Q}_{Y|\mathbf{X}} \right) \right], \tag{2}$$

where $\mathcal{A}(\mathbb{P}^{\text{emp}}) := \{ \mathbb{Q} : \mathbb{Q} \in \mathcal{P}(\mathcal{X} \times \mathcal{Y}) \wedge \mathbb{P}_{\mathbf{X}}^{\text{emp}} = \mathbb{Q}_{\mathbf{X}} \wedge \| \mathbb{E}_{\mathbb{P}^{\text{emp}}} [\boldsymbol{\phi}(\cdot, \cdot)] - \mathbb{E}_{\mathbb{Q}} [\boldsymbol{\phi}(\cdot, \cdot)] \| \leq \varepsilon \}$. The ambiguity set is different from that in Wiesemann et al. (2014) and Farnia & Tse (2016) due to the inequality and feature mapping respectively. The minimization over the function space $\mathcal{H}$ is replaced by directly minimizing over $\mathcal{P}(\mathcal{Y})$ for each $\mathbf{x} \in \mathcal{X}$. The probabilistic predictions are chosen to minimize the worst-case risk evaluated on a set of distributions in an ambiguity set defined by the empirical distribution $\mathbb{P}^{\text{emp}}$ and feature mapping $\boldsymbol{\phi}$. The ambiguity set $\mathcal{A}(\mathbb{P}^{\text{emp}})$ includes distributions that share the same marginal on $\mathcal{X}$ and are no more than $\varepsilon$ away from $\mathbb{P}^{\text{emp}}$ in terms of feature moment divergence. Note that given any feature function $\boldsymbol{\phi}$, the ambiguity set is a compact convex set. Conceptually, we restrict the support of $\mathbb{Q}$ on $\mathcal{X}$ to be the same as the empirical distribution for convenience in both algorithm design and theoretical analysis.

Minimizing the worst-case risk by allowing a certain amount of label uncertainty makes this method inherently robust. It can also be shown to be equivalent to a dual-norm regularized ERM problem:

**Proposition 1** ((Li et al., 2022)). *The distributionally robust probabilistic supervised learning problem based on moment divergence in Eq.* (2) *can be rewritten as*

$$\min_{\boldsymbol{\theta}} \mathbb{E}_{\mathbb{P}_{\mathbf{X}}^{emp}} \underbrace{\min_{\mathbb{P}} \max_{\mathbb{Q}} L\left( \mathbb{P}_{Y|\mathbf{X}}, \mathbb{Q}_{Y|\mathbf{X}} \right) + \boldsymbol{\theta}^{\mathsf{T}} (\mathbb{E}_{\mathbb{Q}_{\check{Y}|\mathbf{X}}} \boldsymbol{\phi}(\boldsymbol{X}, \check{Y}) - \mathbb{E}_{\mathbb{P}_{\check{Y}|\mathbf{X}}^{emp}} \boldsymbol{\phi}(\boldsymbol{X}, \tilde{Y})) + \varepsilon \|\boldsymbol{\theta}\|_*}_{L_{adv}(\boldsymbol{\theta}, \mathbb{P}_{\check{Y}|\mathbf{X}}^{emp})}, \tag{3}$$

*where $\boldsymbol{\theta} \in \mathbb{R}^D$ is the vector of Lagrangian multipliers and $\|\cdot\|_*$ is the dual norm of $\|\cdot\|$.*

We give a proof sketch here. Both $\mathcal{P}(\mathcal{Y})$ and $\mathcal{A}(\tilde{\mathbb{P}})$ are non-empty closed convex sets. Since we assume $L$ is continuous and proper, we know that $L(\cdot, \mathbb{Q})$ is quasi-convex for every $\mathbb{Q}$ and $L(\mathbb{P}, \cdot)$ is

concave for every $\mathbb{P}$ by definition. Eq. (2) is therefore a quasi-convex-concave problem and strong duality holds (Sion, 1958). The regularization is obtained via Lagrangian and Fenchel conjugate.

It is well-known that continuous proper losses are quasi-convex, such as the Brier score, the logarithmic score, the spherical score, the Winkler's score, the ranked probability score, etc. However, some improper (possibly discrete and non-convex) losses can be quasi-convex in the predicted distribution (e.g., the zero-one loss). In contrast, surrogate classification losses are usually convex in a parameter space that is easy to work with, for example, the multiclass hinge loss Weston & Watkins (1998), $\ell_{\mathrm{ww}}(\boldsymbol{\psi}, y) = \sum_{y' \neq y} \max\{0, 1 + \psi_{y'} - \psi_y\}$, and the multiclass logistic loss (Nelder & Wedderburn, 1972), $\ell_{\log}(\boldsymbol{\psi}, y) = \ln\left(\sum_{y'} \exp(\psi_{y'})\right) - \psi_y$, where $\boldsymbol{\psi} \in \mathbb{R}^{|\mathcal{Y}|}$ is a vector of class scores.

From a game theoretic point of view, our formulation in Eq. (2) is equivalent to a two-player zero-sum game in which the predictor player chooses a distribution to minimize the expected game payoff while the adversary player chooses one to maximize the game value while constrained to satisfy certain statistical properties of training data (Grünwald et al., 2004). In the dual problem (Eq. (3)), the Lagrange multipliers parameterize the payoff function for an augmented game and provide a new payoff function for unseen data to construct predictors.

## 3.2 STATISTICAL PROPERTIES

It well known that minimizing strictly proper losses leads to Fisher consistent probability estimation (Williamson et al., 2016). However, minimization of the surrogate risk in Eq. (3) may induce a sub-optimal classifier because of misalignment between the surrogate loss $L_{\mathrm{adv}}$ and the original loss $L$. Fisher consistency provides desirable statistical implications for a surrogate loss such that minimizing it yields an estimator that also minimizes the original loss.

The adversarial surrogate loss $L_{\mathrm{adv}}$ is endowed with an additional regularization term. It reduces to a Fenchel-Young loss (Blondel et al., 2020) when the ambiguity radius $\varepsilon$ is zero. A conclusion of consistency can drawn based on Nowak et al. (2020); Blondel et al. (2020) and our assumption that the groundtruth is probabilistic:

**Corollary 2** ((Li et al., 2022)). *When $\varepsilon = 0$, $L_{adv}$ is Fisher consistent with respect to $L$. Namely, for any $\boldsymbol{x}$,*

$$\mathbb{P}_{Y|\boldsymbol{x}}^{\boldsymbol{\theta}^*_{true}} \in \arg\min_{\mathbb{P}_{Y|\boldsymbol{x}}} L(\mathbb{P}_{Y|\boldsymbol{x}}, \mathbb{P}_{Y|\boldsymbol{x}}^{true})$$

*is the Bayes optimal probabilistic prediction made by $\boldsymbol{\theta}^*_{true}$, the solution in Eq. (3) under $\mathbb{P}^{true}$. The prediction made by $\boldsymbol{\theta}$ is $\mathbb{P}_{Y|\boldsymbol{X}}^{\boldsymbol{\theta}} \in \arg\min_{\mathbb{P}} \max_{\mathbb{Q}} L\left(\mathbb{P}_{Y|\boldsymbol{X}}, \mathbb{Q}_{Y|\boldsymbol{X}}\right) + \mathbb{E}_{\mathbb{Q}_{\check{Y}|\boldsymbol{X}}} \boldsymbol{\theta}^{\intercal} \boldsymbol{\phi}(\boldsymbol{X}, \check{Y})$.*

The consistency result guarantees that the learned probabilistic prediction rules yield Bayes optimal risk as ERM with proper losses in the ideal setting with true distributions and all measurable functions. Also note that the conclusion holds for all quasi-convex losses.

Basic generalization bounds related to true risk for DRO methods can be derived from measure concentration. This approach depends on the choice of ambiguity sets and may have a dimensionality issue. It is also not appropriate for ambiguity sets defined by low-order moments in this paper. Thus, we take an alternate approach following Farnia & Tse (2016) to prove excess out-of-sample risk bounds. We assume $\varepsilon > 0$ to ensure boundedness of $\|\boldsymbol{\theta}\|_*$. We establish the following theorem by making mild assumptions on boundedness on features and losses:

**Theorem 3** ((Li et al., 2022)). *Given $n$ samples, a non-negative multiclass probabilistic loss $L(\cdot, \cdot)$ such that $|L(\cdot, \cdot)| \leq K$, a feature function $\boldsymbol{\phi}(\cdot, \cdot)$ such that $\|\boldsymbol{\phi}(\cdot, \cdot)\| \leq B$ and a positive ambiguity level $\varepsilon > 0$, then, for any $0 < \delta \leq 1$, with a probability at least $1 - \delta$, the following excess true worst-case risk bound holds:*

$$\max_{\mathbb{Q} \in \mathcal{A}(\mathbb{P}^{true})} R_{\mathbb{Q}}^L(\boldsymbol{\theta}^*_{emp}) - \max_{\mathbb{Q} \in \mathcal{A}(\mathbb{P}^{true})} R_{\mathbb{Q}}^L(\boldsymbol{\theta}^*_{true}) \leq \frac{4KB}{\varepsilon\sqrt{n}}\left(1 + \frac{3}{2}\sqrt{\frac{\ln(4/\delta)}{2}}\right), \qquad (4)$$

*where $\boldsymbol{\theta}^*_{emp}$ and $\boldsymbol{\theta}^*_{true}$ are the optimal parameters learned in Eq. (3) under the empirical distribution $\mathbb{P}^{emp}$ and true distribution $\mathbb{P}^{true}$, respectively. The original risk of $\boldsymbol{\theta}$ under $\mathbb{Q}$ is $R_{\mathbb{Q}}^L(\boldsymbol{\theta}) := \mathbb{E}_{\mathbb{Q}_{\boldsymbol{X}, Y}, \mathbb{P}_{Y|\boldsymbol{X}}^{\boldsymbol{\theta}}} L(\mathbb{P}_{Y|\boldsymbol{X}}, \mathbb{Q}_{Y|\boldsymbol{X}})$.*

Theorem 3 improves the results of Asif et al. (2015) and Fathony et al. (2016) that only show qualitative bounds. Under positive regularization, this bound explains the rate of uniform convergence of the true worst-case risk of the estimator $\boldsymbol{\theta}^*_{\text{emp}}$ learned through the empirical distribution $\mathbb{P}^{\text{emp}}$ to the true worst-case risk of the ideal estimator $\boldsymbol{\theta}^*_{\text{true}}$ learned under $\mathbb{P}^{\text{true}}$. Although the empirical estimator is obtained based on a finite set of samples, Theorem 3 justifies the roles which the ambiguity set $\mathcal{A}(\cdot)$, the feature function $\boldsymbol{\phi}(\cdot, \cdot)$, the loss function $L(\cdot, \cdot)$ and the ambiguity parameter $\varepsilon$ play in upper bounding the excess out-of-sample worst-case risk. Intuitively, a larger $\varepsilon$ will reject more hypotheses that are sensitive with larger dual norms, whereas the worst-case risk scales with the range of loss and feature functions.

## 3.3 ALGORITHM

Since $L(\cdot, \cdot)$ is a continuous quasiconvex-concave function, a saddle point in Eq. (3) given $\boldsymbol{\theta}$ must have a zero derivative with respect to $\mathbb{P}$ and $\mathbb{Q}$:

$$\sum_y \mathbb{Q}_{Y|\boldsymbol{x}}(y) \partial \ell(\mathbb{P}_{Y|\boldsymbol{x}}, y)/\partial \mathbb{P}_{Y|\boldsymbol{x}}(y') + Z_{\mathbb{P}_{Y|\boldsymbol{x}}} = 0 \tag{5}$$

$$\ell(\mathbb{P}_{Y|\boldsymbol{x}}, y) + \boldsymbol{\theta}^{\mathsf{T}} \boldsymbol{\phi}(\boldsymbol{x}, y) + Z_{\mathbb{Q}_{Y|\boldsymbol{x}}} = 0, \tag{6}$$

where $Z_{\mathbb{P}_{Y|\boldsymbol{x}}}$ is the Lagrange multipliers for the simplex constraint $\sum_y \mathbb{P}_{Y|\boldsymbol{x}}(y) = 1$, similarly for $Z_{\mathbb{Q}_{Y|\boldsymbol{x}}}$. Note that $Z_{\mathbb{Q}_{Y|\boldsymbol{x}}}$ is constant for all $y$ given $\boldsymbol{x}$. If $\ell$ is local, e.g., $\ell(\mathbb{P}_{Y|\boldsymbol{x}}, y)$ is independent of $\mathbb{P}_{Y|\boldsymbol{x}}(y')$ for $y' \neq y$ and if $\ell(\cdot, y)$ is monotone in $\mathbb{P}_{Y|\boldsymbol{x}}(y) > 0$ (without simplex constraints) with range $\mathbb{R}$, which is the case for the logarithmic loss, Eq. (6) always has a solution and the system of equations for all $y$ along with the simplex constraint $\sum_y \mathbb{P}_{Y|\boldsymbol{x}}(y)$ has a unique solution. With few assumptions on the boundedness of $\ell$ and $\boldsymbol{\theta}^{\mathsf{T}} \boldsymbol{\phi}$, Eq. (6) is ill-posed. Given $\mathbb{P}^*_{Y|\boldsymbol{x}}$ from Eq. (6), the solution $\mathbb{Q}^*_{Y|\boldsymbol{x}}$ to Eq. (5) exists iff

$$\begin{pmatrix} \partial \ell(\mathbb{P}_{Y|\boldsymbol{x}}, 1)/\partial \mathbb{P}_{Y|\boldsymbol{x}}(1) & \dots & \partial \ell(\mathbb{P}_{Y|\boldsymbol{x}}, |\mathcal{Y}|)/\partial \mathbb{P}_{Y|\boldsymbol{x}}(1) & 1 \\ \dots & & & \\ \partial \ell(\mathbb{P}_{Y|\boldsymbol{x}}, 1)/\partial \mathbb{P}_{Y|\boldsymbol{x}}(|\mathcal{Y}|) & \dots & \partial \ell(\mathbb{P}_{Y|\boldsymbol{x}}, |\mathcal{Y}|)/\partial \mathbb{P}_{Y|\boldsymbol{x}}(|\mathcal{Y}|) & 1 \\ 1 & \dots & 1 & 0 \end{pmatrix}$$

is singular. By assuming locality and positiveness, there exists a unique solution $\mathbb{Q}^*_{Y|\boldsymbol{x}}$. One benefit of the proposed method is that users only need to focus on solve Eq. (6) and Eq. (5) for proper losses while Williamson et al. (2016) additionally require a canonical link function for convexity.

Next we show how the system of equations can always be solved with specific losses. We consider an additive combination of the multiclass Brier loss and the logarithmic loss, both of which are continuous strictly proper losses. As indicated by Figure 1, these losses differ primarily in how they penalize the ground truth label's prediction probability as it goes to zero and one. The Brier loss exhibits quadratic growth. The logarithmic loss has a vertical asymptote for labels considered increasingly unlikely to the point of impossibility by the predictor. They have different penalties for underestimation and overestimation of the desired prediction. A trade-off between the log loss and the Brier loss thus provides flexibility to control the cost for misalignment between the prediction and the observation. See appendix for a discussion on including the ranked probability score and other specific losses.

We employ this kind of loss in our DRO method and present an efficient algorithm that can be implemented in practice. With only slight loss of generality and for computational consideration, we assume a fixed positive weight on the log loss. To begin with, the mixture loss is

$$\ell_{\text{mix}}(\mathbb{P}_{Y|\boldsymbol{x}}, y) = -\ln \mathbb{P}_{Y|\boldsymbol{x}}(y) + \beta(1 - 2\mathbb{P}_{Y|\boldsymbol{x}}(y) + \sum_{y'} \mathbb{P}^2_{Y|\boldsymbol{x}}(y')),$$

with derivative

$$\partial \ell_{\text{mix}}(\mathbb{P}_{Y|\boldsymbol{x}}, y)/\partial \mathbb{P}_{Y|\boldsymbol{x}}(y) = -1/\mathbb{P}_{Y|\boldsymbol{x}}(y) - 2\beta + 2\beta \mathbb{P}_{Y|\boldsymbol{x}}(y).$$

Scalar $\beta$ weights the contribution of the Brier loss, to this additive combination, controlling the sensitivity of the predictor to underestimation. The adversarial surrogate of this mixture loss is

Fisher consistent as a direct corollary. Methods that solely mix the predictions of classifiers designed for logarithmic loss minimization and Brier loss optimization, may be appealing for their simplicity, but are demonstrably sub-optimal. For example, with the logistic loss, logistic regression provides a natural parametric form for the predictor, that equates loss minimization with data likelihood maximization.

Although the Brier loss is not local, the additional sum of quadratic terms $\sum_{y'} \mathbb{P}^2_{Y|\boldsymbol{x}}(y')$ is constant across all $y$. Therefore Eq. (6) has a closed form expression in terms of the Lambert $W$ function. Furthermore, the sum over $y$ for all $\mathbb{Q}_{Y|\boldsymbol{x}}(y)$ will cancel out, leaving terms only dependent on the same $y$. So Eq. (5) is simplified into an expression of $\mathbb{Q}$ in terms of $\mathbb{P}$. Normalizing $\mathbb{Q}$ solves $Z_{\mathbb{P}}$, yielding the following proposition:

**Proposition 4.** *The DRO method for a probabilistic loss based on logarithmic loss, and $\beta$ Brier loss has a solution $\mathbb{P}^*_{Y|\mathbf{X}}$ for the predictor parameterized by $\boldsymbol{\theta}$ defined by the following systems of equations:*

$$\forall \mathbf{x} \in \mathcal{X}, \exists C \in \mathbb{R}, \forall y \in \mathcal{Y} \quad \mathbb{P}^*_{Y|\mathbf{x}}(y) = \exp(C + \boldsymbol{\theta}^T \boldsymbol{\phi}(\mathbf{x}, y) - W_0(2\beta e^{C + \boldsymbol{\theta}^T \boldsymbol{\phi}(\mathbf{x}, y)})), \quad (7)$$

*where $C$ is a constant dependent on $\boldsymbol{\theta}$ and $\mathbf{x}$ but independent of $y$, $W(\cdot)$ is the principal branch of the Lambert $W$ function. The corresponding adversary $\mathbb{Q}^*_{Y|\mathbf{X}}$ is defined as*

$$\mathbb{Q}^*_{Y|\mathbf{x}}(y) = \frac{2\beta \mathbb{P}^{*2}_{Y|\mathbf{x}}(y) + Z_{\mathbb{P}_{Y|\mathbf{x}}} \mathbb{P}^*_{Y|\mathbf{x}}(y)}{1 + 2\beta \mathbb{P}^*_{Y|\mathbf{x}}(y)} \; and \; Z_{\mathbb{P}_{Y|\mathbf{x}}} = \frac{1 - \sum_y 2\beta \mathbb{P}^{*2}_{Y|\mathbf{x}}(y)/(1 + 2\beta \hat{\mathbb{P}}^*_{Y|\mathbf{x}}(y))}{\sum_y \mathbb{P}^*_{Y|\mathbf{x}}(y)/(1 + 2\beta \hat{\mathbb{P}}^*_{Y|\mathbf{x}}(y))}.$$
$$(8)$$

Now we show how to solve Eq. (7) with simplex constraints to obtain $\mathbb{P}^*_{Y|\mathbf{x}}$ given $\boldsymbol{\theta}$ for any $\mathbf{x} \in \mathcal{X}$. Let $C = f_y(t) = \boldsymbol{\theta}^T \boldsymbol{\phi}(\mathbf{x}, y) - \ln t - 2\beta t$ be a function of $t = \mathbb{P}^*_{Y|\mathbf{x}}(y)$. By definition, $f(\cdot)$ is a monotonically decreasing function with domain $\mathbb{R}_{++}$ and range $\mathbb{R}$. Its inverse mapping $f^{-1}(\cdot)$ is monotonically decreasing with domain $\mathbb{R}$ and range $\mathbb{R}_{++}$. Therefore, let $g(C) = \sum_y f_y^{-1}(C) = \sum_y \mathbb{P}^*_{Y|\mathbf{x}}(y)$, according to the intermediate value theorem, there exists $C^* \in \mathbb{R}$ such that $g(C^*) = \sum_y \mathbb{P}^*_{Y|\mathbf{x}}(y) = 1$. Because of their monotonicity, we can find $C^*$ and $\mathbb{P}^*_{Y|\mathbf{x}}(\cdot)$ as a solution to Equation

---

**Algorithm 1** Distributionally robust learning for probabilistic supervised learning with a mixture of logistic and Brier losses

**Input:** $\boldsymbol{\phi}, \mathbb{P}^{\text{emp}}_{\mathbf{X},Y}, \beta$, learning rate $\gamma$
**Output:** $\boldsymbol{\theta}^*$
Initialize $\boldsymbol{\theta}$ to be a random vector
**repeat**
    **for all** $\mathbf{x} \in \mathcal{X}$ **do**
        $C, \mathbb{P}^*_{Y|\mathbf{X}}(\cdot|\mathbf{x}) \leftarrow \text{Bisection}(\mathbf{x}, \boldsymbol{\phi}, \boldsymbol{\theta}, \beta)$ by (7)
        Compute $\mathbb{Q}^*_{Y|\mathbf{X}}(\cdot|\mathbf{x})$ by Eq. (8)
    **end for**
    Compute $\partial L_{\text{adv}}/\partial \boldsymbol{\theta}$ by (9)
    $\boldsymbol{\theta} \leftarrow \boldsymbol{\theta} - \gamma \partial L_{\text{adv}}/\partial \boldsymbol{\theta}$
**until** convergence

---

7 via bisection method. Once $\mathbb{P}^*_{Y|\mathbf{X}}$ is obtained, we can find $\mathbb{Q}^*_{Y|\mathbf{X}}$ simply by substitution. After that, the sub-gradient,

$$\partial L_{\text{adv}}/\partial \boldsymbol{\theta} \triangleq \mathbb{E}_{\mathbb{P}^{\text{emp}}_{\mathbf{X}}}(\mathbb{E}_{\mathbb{Q}^*_{Y|\mathbf{X}}}[\boldsymbol{\phi}(\mathbf{X}, Y)] - \mathbb{E}_{\mathbb{P}^{\text{emp}}_{Y|\mathbf{X}}}[\boldsymbol{\phi}(\mathbf{X}, Y)]) + \partial \varepsilon \|\boldsymbol{\theta}\|_*/\partial \boldsymbol{\theta}, \quad (9)$$

can be leveraged to optimize $\boldsymbol{\theta}$. The above steps are summarized in Algorithm 1.

### 3.4 DIFFERENTIABLE LEARNING

By taking advantage of deep neural networks, our method will be able to jointly optimize data representation and the Lagrange multipliers:

$$\min_{\boldsymbol{\theta}, \boldsymbol{\phi}} \mathbb{E}_{\mathbb{P}^{\text{emp}}_{\mathbf{X}}} L_{\text{adv}}(\boldsymbol{\theta}, \mathbb{P}^{\text{emp}}_{\tilde{Y}|\boldsymbol{X}}),$$

enjoying the benefits of end-to-end representation learning without manually looking for a good feature mapping $\boldsymbol{\phi}$. More off-the-shelf mini-batch training tools could be leveraged as well.

We show how to make use of our DRO method as a loss layer in neural network training. A network for supervised learning typically has a linear classification layer in the end without activation. Assume the penultimate layer outputs $\phi(\boldsymbol{x})$, the last layer will output a $|\mathcal{Y}|$-dimensional vector $\boldsymbol{\psi}(\boldsymbol{x}) = [(\boldsymbol{\theta}^{(1)})^{\mathsf{T}}\phi(\boldsymbol{x}), \dots, (\boldsymbol{\theta}^{(|\mathcal{Y}|)})^{\mathsf{T}}\phi(\boldsymbol{x})]$. This is essentially equivalent to adopting a multivector representation to construct $\boldsymbol{\phi}$. Specifically, given $\mathbf{x} \in \mathbb{R}^d$ and $y \in [|\mathcal{Y}|]$, the resulting feature vector $\mathbf{v} = \boldsymbol{\phi}(\mathbf{x}, y) \in \mathbb{R}^{d|\mathcal{Y}|}$ satisfies $v_{yd-d+i} = x_i$ for $i \in [d]$ and $v_j = 0$ otherwise. Therefore taking $\boldsymbol{\psi}(\boldsymbol{x})$ as the input is sufficient for us to compute $\mathbb{P}^*_{Y|\boldsymbol{x}}$ and $\mathbb{Q}^*_{Y|\boldsymbol{x}}$. In this way, our method is the loss layer without learnable parameters, which backpropagates the sub-derivative of loss with respect to $\boldsymbol{\psi}(\boldsymbol{x})$ to the linear classification layer:

$$\mathbb{E}_{\mathbb{P}^{\text{emp}}_{\mathbf{X}}}(\boldsymbol{q}_{Y|\mathbf{X}} - \boldsymbol{p}^{\text{emp}}_{Y|\mathbf{X}}) \in \partial L_{\text{adv}}/\partial \boldsymbol{\psi}(\boldsymbol{x}).$$

Recall $\boldsymbol{q}$ and $\boldsymbol{p}^{\text{emp}}$ are the probability vectors for $\mathbb{Q}$ and $\mathbb{P}^{\text{emp}}$. The sub-gradient with respect to $\boldsymbol{\theta}$ is added to the classification layer.

## 4 EXPERIMENTS

In the experiments, we consider as the performance measure the $L$-risk $R^L_{\mathbb{P}}(h)$, also called the expected generalization loss. The mixture loss $\ell_{\text{mix}}$ of the log loss and Brier loss is adopted. The normalized generalization loss $\frac{1}{(1+\beta)}R^L_{\mathbb{P}^{\text{test}}}(h)$ is estimated based on the test set distribution $\mathbb{P}^{\text{test}}_{\mathbf{X},Y}$.

We compare our adversarial learning approach against neural network models with the softmax and the spherical softmax function as the final normalization layer (Laha et al., 2018). All the baseline methods are able to make use of probabilistic labels in both training and testing. We adopt a three-layer neural network for all the methods, who share the same number of parameters. To make a more fair comparison, we set $\varepsilon = 0$ such that the final classification layer is unregularized. The baselines compute the target loss $L_{\text{mix}}$ based on their probability outputs applied to the logits.

We implement all the methods using PyTorch (Paszke et al., 2019). We use Adam (Kingma & Ba, 2014) for optimization. The number of hidden units is set to $50$. The number of training steps is set to $500$ with a batch size of $64$. We set $\beta = 1$. Default values are used for unmentioned hyperparameters.

We conduct experiments on several real-world datasets, including corel5k (Duygulu et al., 2002), flags (Gonçalves et al., 2013), Stackex_chess (Charte et al., 2015), GpositivePseAAC and GnegativePseAAC (Xu et al., 2016), having statistics reported in Table 1. The ground truth labels in these dataset are either originally probabilistic or converted to a uniform distribution for multi-label classification datasets. At the beginning of each run, we randomly choose $80\%$ of the dataset as the training set and the remaining $20\%$ for evaluation. We further take $20\%$ of the training set as the validation set to determine the best parameter for final testing.

Table 1: Dataset statistics and normalized generalization losses with $95\%$ confidence intervals on each dataset. The best results are indicated in bold. † indicates statistical significance with paired t-test ($p < 0.05$).

| Dataset | corel5k | GnegativePseAAC | flags | GpositivePseAAC | Stackex_chess |
|---|---|---|---|---|---|
| n | 5000 | 1392 | 194 | 519 | 1672 |
| $|\mathcal{Y}|$ | 374 | 8 | 7 | 4 | 227 |
| Features | 499 | 440 | 19 | 440 | 585 |
| Softmax | $2.738 \pm 0.013$ | $0.306 \pm 0.011$ | $\mathbf{1.294 \pm 0.017}$ | $\mathbf{0.329 \pm 0.014}$ | $2.565 \pm 0.031$ |
| Spherical | $2.907 \pm 0.010$† | $0.307 \pm 0.012$ | $1.324 \pm 0.037$ | $0.339 \pm 0.016$† | $2.700 \pm 0.043$† |
| Ours | $\mathbf{2.738 \pm 0.012}$ | $\mathbf{0.306 \pm 0.011}$ | $1.294 \pm 0.017$ | $0.329 \pm 0.014$ | $\mathbf{2.555 \pm 0.037}$ |

We repeat the above process 10 times for each dataset on a laptop with a 2.7 GHz Quad-Core Intel Core i7 CPU. All the methods take less than 1 minute per run in wall time. The results in Table 1 show that our proposed method either has the best performance or achieves similar performance to the best method with no statistical significance in most of the adopted datasets.

For sensitivity analysis, we fix a random split of the Stackex_chess dataset and vary $\beta$ with other settings unchanged. The experiments are repeated 10 times. As shown in Figure 2, the expected loss

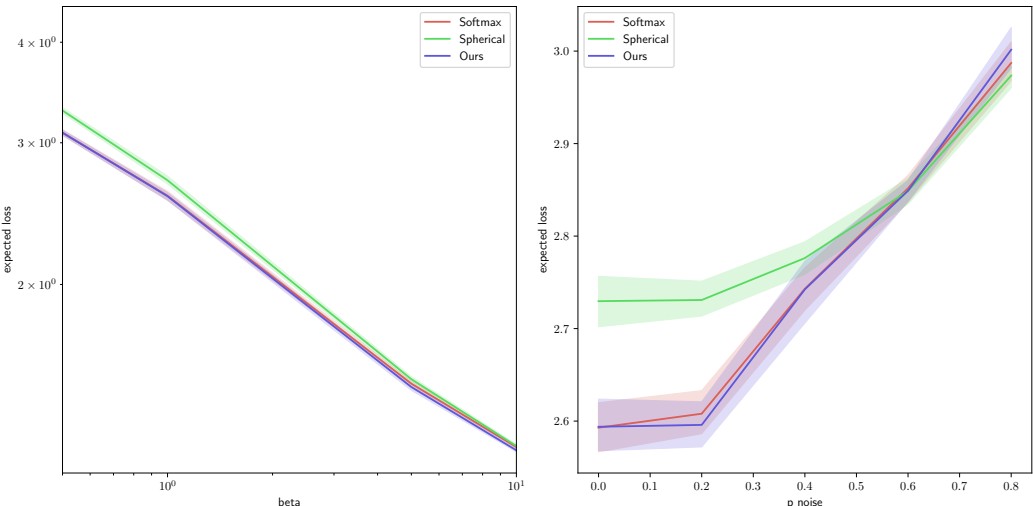

Figure 2: Normalized generalization losses with different coefficients or noise levels. Left: varying $\beta$ in $[0.1, 10.0]$. Right: varying probability of contamination in $[0, 0.8]$. The X axes of the left subfigure is in logarithmic scale. Best viewed in color.

of our method on the test set is slightly better than baselines. For better illustration, we cut $[0.1, 0.5]$ off the x-axis because the softmax and our method are indistinguishable without scaling.

Additionally, we study the robustness of our approach by introducing noise to the training set of the `Stackex_chess` dataset, repeated 10 times. To this end, for each instance $\mathbf{x}$, with a probability $p_{\text{noise}}$, we replace the ground truth by a random distribution from $\mathcal{P}(\mathcal{Y})$. We vary $p_{\text{noise}}$ from 0 to 0.8. As seen in Figure 2, our method is slightly better when $p_{\text{noise}} < 0.4$. All the methods become vulnerable for large $p_{\text{noise}}$ possibly because of the backbone neural network model.

## 5 DISCUSSION AND CONCLUSION

We proposed a moment-based distributionally robust learning framework for probabilistic supervised learning under mild assumptions, showed its equivalence to dual-norm regularization for a surrogate loss, presented its out-of-sample guarantees, developed efficient algorithms for typical continuous proper losses, incorporated the proposed method into differentiable learning and conducted experiments on several real-world datasets. We aimed to shed light on this more general supervised learning setting (Gressmann et al., 2018) and provide a more expressive way of quantifying prediction uncertainty. A drawback of the proposed method is that solving the saddle-point problem can be difficult for some complicated losses while neural networks equipped with a softmax layer makes use of automatic differentiation to avoid facing this issue. Interesting directions for future investigation include generalizing the learning framework to conditional density estimation and considering ambiguity sets defined by higher-order moments.

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

## A  TECHNICAL PROOFS

**Proposition 1.** *The distributionally robust probabilistic supervised learning problem based on moment divergence in Eq.* (2) *can be rewritten as*

$$\min_{\boldsymbol{\theta}} \mathbb{E}_{\mathbb{P}_{\boldsymbol{X}}^{emp}} \underbrace{\min_{\mathbb{P}} \max_{\mathbb{Q}} L\left(\mathbb{P}_{Y|\mathbf{X}}, \mathbb{Q}_{Y|\mathbf{X}}\right) + \boldsymbol{\theta}^{\mathsf{T}}(\mathbb{E}_{\mathbb{Q}_{\check{Y}|\boldsymbol{X}}} \phi(\boldsymbol{X}, \check{Y}) - \mathbb{E}_{\mathbb{P}_{\check{Y}|\boldsymbol{X}}^{emp}} \phi(\boldsymbol{X}, \tilde{Y})) + \varepsilon \|\boldsymbol{\theta}\|_{*}}_{L_{adv}(\boldsymbol{\theta}, \mathbb{P}_{\check{Y}|\boldsymbol{X}}^{emp})},$$

*where* $\boldsymbol{\theta} \in \mathbb{R}^{D}$ *is the vector of Lagrangian multipliers and* $\|\cdot\|_{*}$ *is the dual norm of* $\|\cdot\|$.

*Proof.* Recall the primal problem

$$\min_{\mathbb{P}_{Y|\mathbf{X}} \in \mathcal{P}(\mathcal{Y})} \max_{\mathbb{Q} \in \mathcal{A}(\mathbb{P}^{\text{emp}})} \mathbb{E}_{\mathbb{Q}_{\mathbf{X}}} \left[ L\left(\mathbb{P}_{Y|\mathbf{X}}, \mathbb{Q}_{Y|\mathbf{X}}\right) \right],$$

where $\mathcal{A}(\mathbb{P}^{\text{emp}}) := \{\mathbb{Q} : \mathbb{Q} \in \mathcal{P}(\mathcal{X} \times \mathcal{Y}) \wedge \mathbb{P}_{\mathbf{X}}^{\text{emp}} = \mathbb{Q}_{\mathbf{X}} \wedge \|\mathbb{E}_{\mathbb{P}^{\text{emp}}}\left[\phi(\cdot, \cdot)\right] - \mathbb{E}_{\mathbb{Q}}\left[\phi(\cdot, \cdot)\right]\| \leq \varepsilon\}.$

Note the feature function $\phi(\cdot)$ is fixed and given. The constraint sets $\mathcal{P}(\mathcal{Y})$ and $\mathcal{A}(\mathbb{P}^{\text{emp}})$ are convex. The objective function $L(\mathbb{P}, \mathbb{Q})$ is quasi-convex in $\mathbb{P}$ by (Williamson et al., 2016) and concave in $\mathbb{Q}$ because it is affine in $\mathbb{Q}$. Therefore strong duality holds by Sion's minimax theorem (Sion, 1958):

$$\max_{\mathbb{Q} \in \mathcal{A}(\mathbb{P}^{\text{emp}})} \min_{\mathbb{P}_{Y|\mathbf{X}} \in \mathcal{P}(\mathcal{Y})} \mathbb{E}_{\mathbb{Q}_{\mathbf{X}}} \left[ L\left(\mathbb{P}_{Y|\mathbf{X}}, \mathbb{Q}_{Y|\mathbf{X}}\right) \right].$$

Let $\mathcal{C}(\boldsymbol{u}) := \{\boldsymbol{u} : \|\boldsymbol{u} - \mathbb{E}_{\mathbb{P}^{\text{emp}}}\phi(\cdot)\| \leq \varepsilon\}$. Rewrite the problem with this constraint:

$$\sup_{\mathbb{Q}, \boldsymbol{u}} \min_{\mathbb{P}} \mathbb{E}_{\mathbb{P}_{\mathbf{X}}^{\text{emp}}} \left[ L\left(\mathbb{P}_{Y|\mathbf{X}}, \mathbb{Q}_{Y|\mathbf{X}}\right) \right] - I_{\mathcal{C}}(\boldsymbol{u})$$

$$\text{s.t.} \quad \boldsymbol{u} = \mathbb{E}_{\mathbb{P}_{\mathbf{X}}^{\text{emp}}\mathbb{Q}_{\check{Y}|\mathbf{X}}} \phi(\boldsymbol{X}, \check{Y}),$$

where $I_{\mathcal{C}}(\cdot)$ is the indicator function with $I_{\mathcal{C}}(\boldsymbol{x}) = 0$ if $\boldsymbol{x} \in \mathcal{C}$ and $+\infty$ otherwise. The simplex constraints of $\mathbb{P}$ and $\mathbb{Q}$ are omitted.

The dual problem by relaxing the equality constraint is

$$\sup_{\mathbb{Q}, \boldsymbol{u}} \min_{\boldsymbol{\theta}} \min_{\mathbb{P}} \mathbb{E}_{\mathbb{P}_{\mathbf{X}}^{\text{emp}}} \left[ L\left(\mathbb{P}_{Y|\mathbf{X}}, \mathbb{Q}_{Y|\mathbf{X}}\right) \right] - I_{\mathcal{C}}(\boldsymbol{u}) + \boldsymbol{\theta}^{\mathsf{T}} \mathbb{E}_{\mathbb{P}_{\mathbf{X}}^{\text{emp}}\mathbb{Q}_{\check{Y}|\mathbf{X}}} \phi(\boldsymbol{X}, \check{Y}) - \boldsymbol{\theta}^{\mathsf{T}} \boldsymbol{u},$$

where $\boldsymbol{\theta}$ is the vector of Lagrange multipliers.

Given $\boldsymbol{X} = \boldsymbol{x}$, optimization of $\mathbb{Q}_{\check{Y}|\boldsymbol{x}}$ and $\mathbb{P}_{\hat{Y}|\boldsymbol{x}}$ can be done independently. Again by strong duality, we can rearrange the terms:

$$\min_{\boldsymbol{\theta}} \mathbb{E}_{\mathbb{P}_{\mathbf{X}}^{\text{emp}}} \min_{\mathbb{P}} \max_{\mathbb{Q}} L\left(\mathbb{P}_{Y|\mathbf{X}}, \mathbb{Q}_{Y|\mathbf{X}}\right) + \boldsymbol{\theta}^{\mathsf{T}} \mathbb{E}_{\mathbb{Q}_{\check{Y}|\mathbf{X}}} \phi(\boldsymbol{X}, \check{Y}) + \sup_{\boldsymbol{u}} -I_{\mathcal{C}}(\boldsymbol{u}) - \boldsymbol{\theta}^{\mathsf{T}} \boldsymbol{u}.$$

The associated dual norm $\|\cdot\|_*$ of the norm $\|\cdot\|$ is defined as

$$\|\boldsymbol{z}\|_* := \sup\{\boldsymbol{z}^{\mathsf{T}} \boldsymbol{x} : \|\boldsymbol{x}\| \leq 1\},$$

based on which we are able to simplify the optimization over $\boldsymbol{u}$ as

$$\sup_{\boldsymbol{u}} -I_{\mathcal{C}}(\boldsymbol{u}) - \boldsymbol{\theta}^{\mathsf{T}} \boldsymbol{u} = \sup_{\boldsymbol{u} \in \mathcal{C}} -\boldsymbol{\theta}^{\mathsf{T}} \boldsymbol{u} = \sup_{\boldsymbol{e}:\|\boldsymbol{e}\| \leq 1} -\boldsymbol{\theta}^{\mathsf{T}}(\mathbb{E}_{\mathbb{P}^{\text{emp}}}\phi(\cdot) - \varepsilon \boldsymbol{e}) = -\boldsymbol{\theta}^{\mathsf{T}} \mathbb{E}_{\mathbb{P}^{\text{emp}}}\phi(\cdot) + \varepsilon \|\boldsymbol{\theta}\|_*.$$

Plugging it back to the dual problem, we have

$$\min_{\boldsymbol{\theta}} \mathbb{E}_{\mathbb{P}_{\mathbf{X}}^{\text{emp}}} \min_{\mathbb{P}} \max_{\mathbb{Q}} L\left(\mathbb{P}_{Y|\mathbf{X}}, \mathbb{Q}_{Y|\mathbf{X}}\right) + \boldsymbol{\theta}^{\mathsf{T}}(\mathbb{E}_{\mathbb{Q}_{\check{Y}|\boldsymbol{X}}} \phi(\boldsymbol{X}, \check{Y}) - \mathbb{E}_{\mathbb{P}_{\tilde{Y}|\boldsymbol{X}}^{\text{emp}}} \phi(\boldsymbol{X}, \tilde{Y})) + \varepsilon \|\boldsymbol{\theta}\|_*.$$

$\square$

**Corollary 2.** *When $\varepsilon = 0$, $L_{adv}$ is Fisher consistent with respect to $L$. Namely, for any $\boldsymbol{x}$,*

$$\mathbb{P}_{Y|\boldsymbol{x}}^{\boldsymbol{\theta}_{true}^*} \in \arg\min_{\mathbb{P}_{Y|\boldsymbol{x}}} L(\mathbb{P}_{Y|\boldsymbol{x}}, \mathbb{P}_{Y|\boldsymbol{x}}^{true})$$

*is the Bayes optimal probabilistic prediction made by $\boldsymbol{\theta}_{true}^*$, the solution in Eq. (3) under $\mathbb{P}^{true}$.*

*Proof.* Our setting differs from Nowak et al. (2020) in the fact that we use a distribution as the ground truth. By defining $y^*(\mu)$ as the gold standard probabilistic prediction and $\mathcal{Y}$ as the set of all possible probabilistic predictions in Proposition C.2 in Nowak et al. (2020), we have

$$\mathbb{P}_{\hat{Y}|\boldsymbol{x}}^{\boldsymbol{\theta}_{true}^*} \in \text{Conv}(\arg\min_{\mathbb{P}_{\hat{Y}|\boldsymbol{x}}} L(\mathbb{P}_{Y|\boldsymbol{x}}, \mathbb{P}_{Y|\boldsymbol{x}}^{true})).$$

Because $L$ is assumed continuous proper, any convex combination of minimizers is also a minimizer. Therefore,

$$\mathbb{P}_{\hat{Y}|\boldsymbol{x}}^{\boldsymbol{\theta}_{true}^*} \in \arg\min_{\mathbb{P}_{\hat{Y}|\boldsymbol{x}}} L(\mathbb{P}_{Y|\boldsymbol{x}}, \mathbb{P}_{Y|\boldsymbol{x}}^{true}).$$

$\square$

**Theorem 3.** *Given $n$ samples, a non-negative multiclass probabilistic loss $L(\cdot,\cdot)$ such that $|L(\cdot,\cdot)| \leq K$, a feature function $\phi(\cdot,\cdot)$ such that $\|\phi(\cdot,\cdot)\| \leq B$ and a positive ambiguity level $\varepsilon > 0$, then, for any $0 < \delta \leq 1$, with a probability at least $1 - \delta$, the following excess true worst-case risk bound holds:*

$$\max_{\mathbb{Q}\in\mathcal{A}(\mathbb{P}^{true})} R_{\mathbb{Q}}^{L}(\boldsymbol{\theta}_{emp}^{*}) - \max_{\mathbb{Q}\in\mathcal{A}(\mathbb{P}^{true})} R_{\mathbb{Q}}^{L}(\boldsymbol{\theta}_{true}^{*}) \leq \frac{4KB}{\varepsilon\sqrt{n}}\left(1 + \frac{3}{2}\sqrt{\frac{\ln(4/\delta)}{2}}\right),$$

*where $\boldsymbol{\theta}_{emp}^{*}$ and $\boldsymbol{\theta}_{true}^{*}$ are the optimal parameters learned in Eq. (3) under the empirical distribution $\mathbb{P}^{emp}$ and true distribution $\mathbb{P}^{true}$, respectively. The original risk of $\boldsymbol{\theta}$ under $\mathbb{Q}$ is $R_{\mathbb{Q}}^{L}(\boldsymbol{\theta}) := \mathbb{E}_{\mathbb{Q}_{\boldsymbol{X},Y},\mathbb{P}_{Y|\boldsymbol{X}}^{\boldsymbol{\theta}}} L(\mathbb{P}_{Y|\mathbf{X}},\mathbb{Q}_{Y|\mathbf{X}})$ with prediction $\mathbb{P}_{Y|\boldsymbol{X}}^{\boldsymbol{\theta}} \in \arg\min_{\mathbb{P}}\max_{\mathbb{Q}} L\left(\mathbb{P}_{Y|\mathbf{X}},\mathbb{Q}_{Y|\mathbf{X}}\right) + \mathbb{E}_{\mathbb{Q}_{\check{Y}|\boldsymbol{X}}}\boldsymbol{\theta}^{\mathsf{T}}\phi(\boldsymbol{X},\check{Y})$.*

*Proof.* Define the adversarial surrogate risk of $\boldsymbol{\theta}$ with respect to $\tilde{\mathbb{P}}$ as

$$R_{\tilde{\mathbb{P}}}^{S}(\boldsymbol{\theta}) := \mathbb{E}_{\tilde{\mathbb{P}}_{\boldsymbol{X}}}\min_{\mathbb{P}}\max_{\mathbb{Q}} L\left(\mathbb{P}_{Y|\mathbf{X}},\mathbb{Q}_{Y|\mathbf{X}}\right) + \boldsymbol{\theta}^{\mathsf{T}}(\mathbb{E}_{\mathbb{Q}_{\check{Y}|\boldsymbol{X}}}\phi(\boldsymbol{X},\check{Y}) - \mathbb{E}_{\tilde{\mathbb{P}}_{\tilde{Y}|\boldsymbol{X}}}\phi(\boldsymbol{X},\tilde{Y})) + \varepsilon\|\boldsymbol{\theta}\|_{*}.$$

Let $\boldsymbol{\theta}_{\text{true}}^{*} \in \arg\min_{\boldsymbol{\theta}} R_{\mathbb{P}^{\text{true}}}^{S}(\boldsymbol{\theta})$ and $\boldsymbol{\theta}_{\text{emp}}^{*} \in \arg\min_{\boldsymbol{\theta}} R_{\mathbb{P}^{\text{emp}}}^{S}(\boldsymbol{\theta})$ be the optimal parameters learned with $\mathbb{P}_{\mathbf{X},Y}^{\text{true}}$ and $\mathbb{P}_{\mathbf{X},Y}^{\text{emp}}$ respectively.

Given $\boldsymbol{x}$, define the decoded prediction by $\boldsymbol{\theta}$ as

$$\mathbb{P}_{Y|\boldsymbol{x}}^{\boldsymbol{\theta}} \in \arg\min_{\mathbb{P}}\max_{\mathbb{Q}} L\left(\mathbb{P}_{Y|\mathbf{X}},\mathbb{Q}_{Y|\mathbf{X}}\right) + \boldsymbol{\theta}^{\mathsf{T}}\mathbb{E}_{\mathbb{Q}_{\check{Y}|\boldsymbol{x}}}\phi(\boldsymbol{X},\check{Y}).$$

Let the original risk of loss $L$ under some distribution $\mathbb{Q}$ be

$$R_{\mathbb{Q}}^{L}(\boldsymbol{\theta}) := \mathbb{E}_{\mathbb{Q}_{\boldsymbol{x}}} L\left(\mathbb{P}_{Y|\mathbf{X}}^{\boldsymbol{\theta}},\mathbb{Q}_{Y|\mathbf{X}}\right).$$

According to Proposition 1, for any fixed $\mathbb{P}$, we have similarly

$$\max_{\mathbb{Q}\in\mathcal{A}(\mathbb{P}^{\text{emp}})} \mathbb{E}_{\mathbb{Q}_{\boldsymbol{x}}} L\left(\mathbb{P}_{Y|\mathbf{X}}^{\boldsymbol{\theta}},\mathbb{Q}_{Y|\mathbf{X}}\right) \triangleq \min_{\boldsymbol{\theta}} \mathbb{E}_{\mathbb{P}_{\boldsymbol{X}}^{\text{emp}}}\max_{\mathbb{Q}} L\left(\mathbb{P}_{Y|\mathbf{X}},\mathbb{Q}_{Y|\mathbf{X}}\right) + \boldsymbol{\theta}^{\mathsf{T}}(\mathbb{E}_{\mathbb{Q}_{\check{Y}|\boldsymbol{x}}}\phi(\boldsymbol{X},\check{Y}) - \mathbb{E}_{\mathbb{P}_{\tilde{Y}|\boldsymbol{x}}^{\text{emp}}}\phi(\boldsymbol{X},\tilde{Y})) + \varepsilon\|\boldsymbol{\theta}\|_{*}.$$

We start by looking at the worst-case risk of $\boldsymbol{\theta}_{\text{true}}^{*}$ and $\boldsymbol{\theta}_{\text{emp}}^{*}$.

$$\max_{\mathbb{Q}\in\mathcal{A}(\mathbb{P}^{\text{true}})} R_{\mathbb{Q}}^{L}(\boldsymbol{\theta}_{\text{emp}}^{*})$$

$$= \min_{\boldsymbol{\theta}} \mathbb{E}_{\mathbb{P}_{\boldsymbol{X}}^{\text{true}}}\max_{\mathbb{Q}} L\left(\mathbb{P}_{Y|\mathbf{X}}^{\boldsymbol{\theta}_{\text{emp}}^{*}},\mathbb{Q}_{Y|\mathbf{X}}\right) + \boldsymbol{\theta}^{\mathsf{T}}(\mathbb{E}_{\mathbb{Q}_{\check{Y}|\boldsymbol{x}}}\phi(\boldsymbol{X},\check{Y}) - \mathbb{E}_{\mathbb{P}_{Y|\boldsymbol{x}}^{\text{true}}}\phi(\boldsymbol{X},Y)) + \varepsilon\|\boldsymbol{\theta}\|_{*}$$

$$\leq \mathbb{E}_{\mathbb{P}_{\boldsymbol{X}}^{\text{true}}}\max_{\mathbb{Q}} L\left(\mathbb{P}_{Y|\mathbf{X}}^{\boldsymbol{\theta}_{\text{emp}}^{*}},\mathbb{Q}_{Y|\mathbf{X}}\right) + \boldsymbol{\theta}_{\text{emp}}^{*}\cdot(\mathbb{E}_{\mathbb{Q}_{\check{Y}|\boldsymbol{x}}}\phi(\boldsymbol{X},\check{Y}) - \mathbb{E}_{\mathbb{P}_{Y|\boldsymbol{x}}^{\text{true}}}\phi(\boldsymbol{X},Y)) + \varepsilon\|\boldsymbol{\theta}_{\text{emp}}^{*}\|_{*},$$

where the last inequality holds because $\boldsymbol{\theta}_{\text{emp}}^{*}$ is not necessarily a minimizer. Similarly for $\boldsymbol{\theta}_{\text{true}}^{*}$,

$$\max_{\mathbb{Q}\in\mathcal{A}(\mathbb{P}^{\text{true}})} R_{\mathbb{Q}}^{L}(\boldsymbol{\theta}_{\text{true}}^{*}) \leq \mathbb{E}_{\mathbb{P}_{\boldsymbol{X}}^{\text{true}}}\max_{\mathbb{Q}} L\left(\mathbb{P}_{Y|\mathbf{X}}^{\boldsymbol{\theta}_{\text{true}}^{*}},\mathbb{Q}_{Y|\mathbf{X}}\right) + \boldsymbol{\theta}_{\text{true}}^{*}\cdot(\mathbb{E}_{\mathbb{Q}_{\check{Y}|\boldsymbol{x}}}\phi(\boldsymbol{X},\check{Y}) - \mathbb{E}_{\mathbb{P}_{Y|\boldsymbol{x}}^{\text{true}}}\phi(\boldsymbol{X},Y)) + \varepsilon\|\boldsymbol{\theta}_{\text{true}}^{*}\|_{*}.$$

On the other hand,

$$\mathbb{E}_{\mathbb{P}_{\boldsymbol{X}}^{\text{true}}}\max_{\mathbb{Q}} L\left(\mathbb{P}_{Y|\mathbf{X}}^{\boldsymbol{\theta}_{\text{true}}^{*}},\mathbb{Q}_{Y|\mathbf{X}}\right) + \boldsymbol{\theta}_{\text{true}}^{*}\cdot(\mathbb{E}_{\mathbb{Q}_{\check{Y}|\boldsymbol{x}}}\phi(\boldsymbol{X},\check{Y}) - \mathbb{E}_{\mathbb{P}_{Y|\boldsymbol{x}}^{\text{true}}}\phi(\boldsymbol{X},Y)) + \varepsilon\|\boldsymbol{\theta}_{\text{true}}^{*}\|_{*}$$

$$= \min_{\boldsymbol{\theta}} \mathbb{E}_{\mathbb{P}_{\boldsymbol{X}}^{\text{true}}}\min_{\mathbb{P}}\max_{\mathbb{Q}} L\left(\mathbb{P}_{Y|\mathbf{X}},\mathbb{Q}_{Y|\mathbf{X}}\right) + \boldsymbol{\theta}^{\mathsf{T}}(\mathbb{E}_{\mathbb{Q}_{\check{Y}|\boldsymbol{x}}}\phi(\boldsymbol{X},\check{Y}) - \mathbb{E}_{\mathbb{P}_{Y|\boldsymbol{x}}^{\text{true}}}\phi(\boldsymbol{X},Y)) + \varepsilon\|\boldsymbol{\theta}\|_{*}$$

$$= \min_{\mathbb{P}}\min_{\boldsymbol{\theta}} \mathbb{E}_{\mathbb{P}_{\boldsymbol{X}}^{\text{true}}}\max_{\mathbb{Q}} L\left(\mathbb{P}_{Y|\mathbf{X}},\mathbb{Q}_{Y|\mathbf{X}}\right) + \boldsymbol{\theta}^{\mathsf{T}}(\mathbb{E}_{\mathbb{Q}_{\check{Y}|\boldsymbol{x}}}\phi(\boldsymbol{X},\check{Y}) - \mathbb{E}_{\mathbb{P}_{Y|\boldsymbol{x}}^{\text{true}}}\phi(\boldsymbol{X},Y)) + \varepsilon\|\boldsymbol{\theta}\|_{*}$$

$$\leq \min_{\boldsymbol{\theta}} \mathbb{E}_{\mathbb{P}_{\boldsymbol{X}}^{\text{true}}}\max_{\mathbb{Q}} L\left(\mathbb{P}_{Y|\mathbf{X}}^{\boldsymbol{\theta}_{\text{true}}^{*}},\mathbb{Q}_{Y|\mathbf{X}}\right) + \boldsymbol{\theta}^{\mathsf{T}}(\mathbb{E}_{\mathbb{Q}_{\check{Y}|\boldsymbol{x}}}\phi(\boldsymbol{X},\check{Y}) - \mathbb{E}_{\mathbb{P}_{Y|\boldsymbol{x}}^{\text{true}}}\phi(\boldsymbol{X},Y)) + \varepsilon\|\boldsymbol{\theta}\|_{*}$$

$$= \max_{\mathbb{Q}\in\mathcal{A}(\mathbb{P}^{\text{true}})} R_{\mathbb{Q}}^{L}(\boldsymbol{\theta}_{\text{true}}^{*}),$$

where the first equality holds according to the definition of $\boldsymbol{\theta}^*_{\text{true}}$. The above two inequalities imply the equality:

$$\max_{\mathbb{Q}\in\mathcal{A}(\mathbb{P}^{\text{true}})} R^L_{\mathbb{Q}}(\boldsymbol{\theta}^*_{\text{true}}) = \mathbb{E}_{\mathbb{P}^{\text{true}}_{\boldsymbol{X}}} \max_{\mathbb{Q}} L\left(\mathbb{P}^{\boldsymbol{\theta}^*_{\text{true}}}_{Y|\mathbf{X}}, \mathbb{Q}_{Y|\mathbf{X}}\right) + \boldsymbol{\theta}^*_{\text{true}}\cdot(\mathbb{E}_{\mathbb{Q}_{\check{Y}|\boldsymbol{x}}}\boldsymbol{\phi}(\boldsymbol{X},\check{Y}) - \mathbb{E}_{\mathbb{P}^{\text{true}}_{Y|\boldsymbol{x}}}\boldsymbol{\phi}(\boldsymbol{X},Y)) + \varepsilon\|\boldsymbol{\theta}^*_{\text{true}}\|_*.$$

Therefore,

$$\max_{\mathbb{Q}\in\mathcal{A}(\mathbb{P}^{\text{true}})} R^L_{\mathbb{Q}}(\boldsymbol{\theta}^*_{\text{emp}}) - \max_{\mathbb{Q}\in\mathcal{A}(\mathbb{P}^{\text{true}})} R^L_{\mathbb{Q}}(\boldsymbol{\theta}^*_{\text{true}})$$

$$\leq (\mathbb{E}_{\mathbb{P}^{\text{true}}_{\boldsymbol{X}}} \max_{\mathbb{Q}} L\left(\mathbb{P}^{\boldsymbol{\theta}^*_{\text{true}}}_{Y|\mathbf{X}}, \mathbb{Q}_{Y|\mathbf{X}}\right) + \boldsymbol{\theta}^*_{\text{emp}}\cdot(\mathbb{E}_{\mathbb{Q}_{\check{Y}|\boldsymbol{x}}}\boldsymbol{\phi}(\boldsymbol{X},\check{Y}) - \mathbb{E}_{\mathbb{P}^{\text{true}}_{Y|\boldsymbol{x}}}\boldsymbol{\phi}(\boldsymbol{X},Y)) + \varepsilon\|\boldsymbol{\theta}^*_{\text{emp}}\|_*)$$

$$- (\mathbb{E}_{\mathbb{P}^{\text{true}}_{\boldsymbol{X}}} \max_{\mathbb{Q}} L\left(\mathbb{P}^{\boldsymbol{\theta}^*_{\text{true}}}_{Y|\mathbf{X}}, \mathbb{Q}_{Y|\mathbf{X}}\right) + \boldsymbol{\theta}^*_{\text{true}}\cdot(\mathbb{E}_{\mathbb{Q}_{\check{Y}|\boldsymbol{x}}}\boldsymbol{\phi}(\boldsymbol{X},\check{Y}) - \mathbb{E}_{\mathbb{P}^{\text{true}}_{Y|\boldsymbol{x}}}\boldsymbol{\phi}(\boldsymbol{X},Y)) + \varepsilon\|\boldsymbol{\theta}^*_{\text{true}}\|_*).$$

$$(10)$$

The main idea is thus to use uniform convergence bound. Firstly, by substituting $\mathbb{Q} = \mathbb{P}^{\text{true}}$, note that

$$\min_{\mathbb{P}} \max_{\mathbb{Q}} L\left(\mathbb{P}_{Y|\mathbf{X}}, \mathbb{Q}_{Y|\mathbf{X}}\right) + \boldsymbol{\theta}^{\mathsf{T}}(\mathbb{E}_{\mathbb{Q}_{\check{Y}|\boldsymbol{x}}}\boldsymbol{\phi}(\boldsymbol{X},\check{Y}) - \mathbb{E}_{\mathbb{P}^{\text{true}}_{Y|\boldsymbol{x}}}\boldsymbol{\phi}(\boldsymbol{X},Y)) \geq \min_{\mathbb{P}} L\left(\mathbb{P}_{Y|\mathbf{X}}, \mathbb{P}^{\text{true}}_{Y|\mathbf{X}}\right) \geq 0.$$

We can get an upper bound of the norm of any optimal solution $\boldsymbol{\theta}^*_{\text{true}}$ or $\boldsymbol{\theta}^*_{\text{emp}}$ as follows:

$$0 + \varepsilon\|\boldsymbol{\theta}^*_{\text{true}}\|_* \leq R^S_{\mathbb{P}^{\text{true}}}(\boldsymbol{\theta}^*_{\text{true}}) \leq R^S_{\mathbb{P}^{\text{true}}}(\mathbf{0}) \leq \mathbb{E}_{\mathbb{P}^{\text{true}}_{\boldsymbol{X}}} L\left(\mathbb{P}_{Y|\mathbf{X}}, \mathbb{Q}_{Y|\mathbf{X}}\right) \leq K \implies \|\boldsymbol{\theta}^*_{\text{true}}\|_* \leq \frac{K}{\varepsilon}.$$

Let $\psi(\boldsymbol{X},Y) := \boldsymbol{\theta}^{\mathsf{T}}\boldsymbol{\phi}(\boldsymbol{X},Y)$ and $\boldsymbol{\psi}_{\boldsymbol{x}} := (\psi(\boldsymbol{x},Y))_{Y\in\mathcal{Y}}$. Define

$$f(\boldsymbol{\theta},\tilde{\mathbb{P}}) := \mathbb{E}_{\tilde{\mathbb{P}}_{\boldsymbol{X}}} \min_{\mathbb{P}} \max_{\mathbb{Q}} L\left(\mathbb{P}_{Y|\mathbf{X}}, \mathbb{Q}_{Y|\mathbf{X}}\right) + \boldsymbol{\theta}^{\mathsf{T}}(\mathbb{E}_{\mathbb{Q}_{\check{Y}|\boldsymbol{x}}}\boldsymbol{\phi}(\boldsymbol{X},\check{Y}) - \mathbb{E}_{\tilde{\mathbb{P}}_{\tilde{Y}|\boldsymbol{x}}}\boldsymbol{\phi}(\boldsymbol{X},\tilde{Y}))$$

$$\triangleq \mathbb{E}_{\tilde{\mathbb{P}}_{\boldsymbol{X}}} \max_{\mathbb{Q}} L\left(\mathbb{P}^{\boldsymbol{\theta}}_{Y|\mathbf{X}}, \mathbb{Q}_{Y|\mathbf{X}}\right) + \boldsymbol{\theta}^{\mathsf{T}}(\mathbb{E}_{\mathbb{Q}_{\check{Y}|\boldsymbol{x}}}\boldsymbol{\phi}(\boldsymbol{X},\check{Y}) - \mathbb{E}_{\tilde{\mathbb{P}}_{\tilde{Y}|\boldsymbol{x}}}\boldsymbol{\phi}(\boldsymbol{X},\tilde{Y}))$$

$$\triangleq \mathbb{E}_{\tilde{\mathbb{P}}_{\boldsymbol{X}}} \max_{\mathbb{Q}} L\left(\mathbb{P}^{\boldsymbol{\theta}}_{Y|\mathbf{X}}, \mathbb{Q}_{Y|\mathbf{X}}\right) + (\mathbb{E}_{\mathbb{Q}_{\check{Y}|\boldsymbol{x}}}\psi(\boldsymbol{X},\check{Y}) - \mathbb{E}_{\tilde{\mathbb{P}}_{\tilde{Y}|\boldsymbol{x}}}\psi(\boldsymbol{X},\tilde{Y}))$$

$$\triangleq g(\boldsymbol{\psi},\tilde{\mathbb{P}}).$$

Let $\boldsymbol{q}_{\boldsymbol{x}} \in \Delta$ be the probability vector of $\mathbb{Q}_{\check{Y}|\boldsymbol{x}}$ and $\boldsymbol{e}_Y$ be the standard basis vector with $Y$-th entry equal to 1. We have that for any $(\boldsymbol{x},Y)$,

$$\frac{\partial}{\partial\boldsymbol{\psi}_{\boldsymbol{x}}} g(\boldsymbol{\psi},\delta_{(\boldsymbol{x},Y)}) \subseteq \text{Conv}(\{\boldsymbol{q}_{\boldsymbol{x}} - \boldsymbol{e}_Y : \boldsymbol{q}_{\boldsymbol{x}} \in \Delta\}) \implies \|\frac{\partial}{\partial\boldsymbol{\psi}_{\boldsymbol{x}}} g(\boldsymbol{\psi},\delta_{(\boldsymbol{x},Y)})\|_1 \leq \max_{\boldsymbol{q}_{\boldsymbol{x}}\in\Delta}\|\boldsymbol{q}_{\boldsymbol{x}} - \boldsymbol{e}_Y\|_1 \leq 2,$$

where $\delta_{(\boldsymbol{x},Y)}$ is the Dirac point measure. $g(\boldsymbol{\psi},\tilde{\mathbb{P}})$ is therefore 2-Lipschitz with respect to the $\ell_1$ norm. As per the assumption, $\|\boldsymbol{\phi}(\cdot,\cdot)\| \leq B$. This further implies that

$$f(\boldsymbol{\theta}_1,\delta_{(\boldsymbol{x}_1,Y_1)}) - f(\boldsymbol{\theta}_2,\delta_{(\boldsymbol{x}_2,Y_2)}) \leq \frac{4KB}{\varepsilon} \quad \forall\boldsymbol{\theta}_1,\boldsymbol{\theta}_2,\boldsymbol{x}_1,\boldsymbol{x}_2,Y_1,Y_2 \quad \text{s.t.} \quad \|\boldsymbol{\theta}_i\|_* \leq \frac{K}{\varepsilon} \quad \forall i = 1, 2.$$

We then follow the proof of Theorem 3 in Farnia & Tse (2016). According to Theorem 26.12 in Shalev-Shwartz & Ben-David (2014), by uniform convergence, for any $\delta \in (0, 2]$, with a probability at least $1 - \frac{\delta}{2}$,

$$f(\boldsymbol{\theta}^*_{\text{emp}}, \mathbb{P}^{\text{true}}) - f(\boldsymbol{\theta}^*_{\text{emp}}, \mathbb{P}^{\text{emp}}) \leq \frac{4KB}{\varepsilon\sqrt{n}}\left(1 + \sqrt{\frac{\ln(4/\delta)}{2}}\right).$$

According to the definition of $\boldsymbol{\theta}^*_{\text{true}}$, the following inequality holds:

$$f(\boldsymbol{\theta}^*_{\text{emp}}, \mathbb{P}^{\text{emp}}) + \varepsilon\|\boldsymbol{\theta}^*_{\text{emp}}\|_* - f(\boldsymbol{\theta}^*_{\text{true}}, \mathbb{P}^{\text{emp}}) - \varepsilon\|\boldsymbol{\theta}^*_{\text{true}}\|_* \leq 0.$$

Since $\boldsymbol{\theta}_{\text{true}}^*$ do not depend on samples, according to the Hoeffding's inequality, with a probability $1 - \delta/2$,

$$f(\boldsymbol{\theta}_{\text{true}}^*, \mathbb{P}^{\text{emp}}) - f(\boldsymbol{\theta}_{\text{true}}^*, \mathbb{P}^{\text{true}}) \leq \frac{2KB}{\varepsilon\sqrt{n}} \sqrt{\frac{\ln(4/\delta)}{2}}.$$

Applying the union bound to the above three inequations, with a probability $1 - \delta$, we have

$$f(\boldsymbol{\theta}_{\text{emp}}^*, \mathbb{P}^{\text{true}}) + \varepsilon\|\boldsymbol{\theta}_{\text{emp}}^*\|_* - f(\boldsymbol{\theta}_{\text{true}}^*, \mathbb{P}^{\text{true}}) - \varepsilon\|\boldsymbol{\theta}_{\text{true}}^*\|_* \leq \frac{4KB}{\varepsilon\sqrt{n}} \left(1 + \frac{3}{2}\sqrt{\frac{\ln(4/\delta)}{2}}\right).$$

As stated by Inequation (10), we conclude with the following excess risk bound:

$$\max_{\mathbb{Q}\in\mathcal{A}(\mathbb{P}^{\text{true}})} R_\mathbb{Q}^L(\boldsymbol{\theta}_{\text{emp}}^*) - \max_{\mathbb{Q}\in\mathcal{A}(\mathbb{P}^{\text{true}})} R_\mathbb{Q}^L(\boldsymbol{\theta}_{\text{true}}^*) \leq \frac{4KB}{\varepsilon\sqrt{n}} \left(1 + \frac{3}{2}\sqrt{\frac{\ln(4/\delta)}{2}}\right).$$

$\square$

**Proposition 4.** *The DRO method for a probabilistic loss based on logarithmic loss, and $\beta$ Brier loss has a solution $\mathbb{P}_{Y|\mathbf{X}}^*$ for the predictor parameterized by $\boldsymbol{\theta}$ defined by the following systems of equations:*

$$\forall \mathbf{x}\in\mathcal{X}, \exists C\in\mathbb{R}, \forall y\in\mathcal{Y} \quad \mathbb{P}_{Y|\mathbf{x}}^*(y) = \exp(C + \boldsymbol{\theta}^T\phi(\mathbf{x}, y) - W_0(2\beta e^{C+\boldsymbol{\theta}^T\phi(\mathbf{x},y)})),$$

*where $C$ is a constant dependent on $\boldsymbol{\theta}$ and $\mathbf{x}$ but independent of $y$, $W(\cdot)$ is the principal branch of the Lambert W function. The corresponding adversary $\mathbb{Q}_{Y|\mathbf{X}}^*$ is defined as*

$$\mathbb{Q}_{Y|\mathbf{x}}^*(y) = \frac{2\beta\mathbb{P}_{Y|\mathbf{x}}^{*2}(y) + Z_{\mathbb{P}_{Y|\mathbf{x}}}\mathbb{P}_{Y|\mathbf{x}}^*(y)}{1 + 2\beta\mathbb{P}_{Y|\mathbf{x}}^*(y)} \quad and \quad Z_{\mathbb{P}_{Y|\mathbf{x}}} = \frac{1 - \sum_y 2\beta\mathbb{P}_{Y|\mathbf{x}}^{*2}(y)/(1 + 2\beta\hat{\mathbb{P}}_{Y|\mathbf{x}}^*(y))}{\sum_y \mathbb{P}_{Y|\mathbf{x}}^*(y)/(1 + 2\beta\hat{\mathbb{P}}_{Y|\mathbf{x}}^*(y))}.$$

*Proof.* Recall the saddle-point optimality condition:

$$\sum_y \mathbb{Q}_Y(y)\partial\ell(\mathbb{P}_Y, y)/\partial\mathbb{P}_Y(y') + Z_{\mathbb{P}_Y} = 0$$

$$\ell(\mathbb{P}_Y, y) + \boldsymbol{\theta}^\mathsf{T}\phi(\boldsymbol{x}, y) + Z_{\mathbb{Q}_Y} = 0.$$

Dependence on $\boldsymbol{x}$ is omitted when context is clear. Substituting $\ell_{\text{mix}}$ yields:

$$\mathbb{Q}_Y(y)(-\frac{1}{\mathbb{P}_Y(y)} - 2\beta) + 2\beta\mathbb{P}_Y(y) + Z_{\mathbb{P}_Y} = 0$$

$$-\ln\mathbb{P}_Y(y) + \beta(1 - 2\mathbb{P}_Y(y) + \sum_{y'}\mathbb{P}_Y^2(y')) + \boldsymbol{\theta}^\mathsf{T}\phi(\boldsymbol{x}, y) + Z_{\mathbb{Q}_Y} = 0.$$

Note that $C := \beta + \beta\sum_{y'}\mathbb{P}_Y^2(y') + Z_{\mathbb{Q}_Y}$ is constant across all $y$'s given $\boldsymbol{\theta}$, $\boldsymbol{x}$. Thus for fixed $\boldsymbol{\theta}$, $\boldsymbol{x}$, we have for some $C_{\boldsymbol{\theta},\boldsymbol{x}}^*$,

$$C_{\boldsymbol{\theta},\boldsymbol{x}}^* + \boldsymbol{\theta}\cdot\phi(\boldsymbol{x}, y) = \ln\mathbb{P}_Y(y) + 2\beta\mathbb{P}_Y(y) \quad \forall y\in\mathcal{Y},$$

which is equivalent to

$$2\beta\mathbb{P}_Y(y)e^{2\beta\mathbb{P}_Y(y)} = 2\beta e^{\boldsymbol{\theta}\cdot\phi(\boldsymbol{x},y)+C_{\boldsymbol{\theta},\boldsymbol{x}}^*}.$$

By the definition of the Lambert $W$ function,

$$2\beta\mathbb{P}_Y(y) = W(2\beta e^{\boldsymbol{\theta}\cdot\phi(\boldsymbol{x},y)+C_{\boldsymbol{\theta},\boldsymbol{x}}^*}).$$

Since $2\beta e^{\boldsymbol{\theta}\cdot\phi(\boldsymbol{x},y)+C_{\boldsymbol{\theta},\boldsymbol{x}}^*} \geq 0$, the principal branch $W_0$ of the Lamber $W$ function is always applicable. Also by the formula $e^{-W(x)} = \frac{W(x)}{x}$, we have

$$\mathbb{P}_Y(y) = \exp(C_{\boldsymbol{\theta},\boldsymbol{x}}^* + \boldsymbol{\theta}^T\phi(\boldsymbol{x}, y) - W_0(2\beta e^{C_{\boldsymbol{\theta},\boldsymbol{x}}^*+\boldsymbol{\theta}^T\phi(\mathbf{x},y)})) \quad \forall y.$$

Let $\mathbb{P}_Y^*$ (for a given $\boldsymbol{\theta}$) be a solution to this set of equations that also satisfies $\sum_y \mathbb{P}_Y^*(y) = 1$. By Eq. (5), the optimal $\mathbb{Q}$ satisfies

$$\mathbb{Q}_Y^*(y) = \frac{2\beta\mathbb{P}_Y^*(y) + Z_{\mathbb{P}_Y}}{\frac{1}{\mathbb{P}_Y^*(y)} + 2\beta} = \frac{2\beta\mathbb{P}_Y^{*2}(y) + Z_{\mathbb{P}_Y}\mathbb{P}_Y^*(y)}{1 + 2\beta\mathbb{P}_Y^*(y)}.$$

$Z_{\mathbb{P}_Y}$ must be chosen to properly normalize $\mathbb{Q}_Y^*(y)$:

$$\sum_y \mathbb{Q}_Y^*(y) = Z_{\mathbb{P}_Y}\sum_y \frac{1}{\frac{1}{\mathbb{P}_Y^*(y)} + \alpha + 2\beta} + \sum_y \frac{2\beta\mathbb{P}_Y^*(y)}{\frac{1}{\mathbb{P}_Y^*(y)} + \alpha + 2\beta} = 1$$

$$\implies Z_{\mathbb{P}_Y}^* = \frac{1 - \sum_y \frac{2\beta\mathbb{P}_Y^*(y)}{\frac{1}{\mathbb{P}_Y^*(y)} + \alpha + 2\beta}}{\sum_y \frac{1}{\frac{1}{\mathbb{P}_Y^*(y)} + \alpha + 2\beta}} = \frac{1 - \sum_y \frac{2\beta\mathbb{P}_Y^{*2}(y)}{1 + (\alpha + 2\beta)\mathbb{P}_Y^*(y)}}{\sum_y \frac{\mathbb{P}_Y^*(y)}{1 + (\alpha + 2\beta)\mathbb{P}_Y^*(y)}}.$$

Both $Z_{\mathbb{P}_Y}^*$ and $\mathbb{Q}_Y^*(y)$ are positive because $\mathbb{P}_Y^* \in \mathcal{P}(\mathcal{Y})$ is a solution. $\qquad\square$

## B  MORE LOSSES

The discrete ranked probability vector assumes an ordering relationship in $\mathcal{Y}$, i.e., $\mathcal{Y} := \{1, 2, \ldots, |\mathcal{Y}|\}$. The score can be written as

$$\ell_{\mathrm{rp}}(\mathbb{P}_Y, y) := \sum_{i=1}^{|\mathcal{Y}|}[\sum_{j=1}^{i}\mathbb{P}_Y(j) - \mathbb{I}(i \geq y)]^2.$$

The mixture loss of the log loss, Brier loss and ranked probability loss can be written as

$$\ell_{\mathrm{mix}}(\mathbb{P}_Y, y) = -\ln\mathbb{P}_Y(y) + \beta(1 - 2\mathbb{P}_Y(y) + \sum_{y'}\mathbb{P}_Y^2(y')) + \alpha\sum_{i=1}^{|\mathcal{Y}|}[\sum_{j=1}^{i}\mathbb{P}_Y(j) - \mathbb{I}(i \geq y)]^2.$$

Substituting the loss into Eq. (6) yields

$$\mathbb{Q}_Y(y)(-\frac{1}{\mathbb{P}_Y(y)} - 2\beta) + 2\beta\mathbb{P}_Y(y) + 2\alpha\sum_{i=y}^{|\mathcal{Y}|}\sum_{j=1}^{i}\mathbb{P}_Y(j) + Z_{\mathbb{P}_Y} - 2\alpha(|\mathcal{Y}| - y + 1 - \sum_{i=y+1}^{|\mathcal{Y}|}(i - y)\mathbb{Q}_Y(i)) = 0 \tag{11}$$

$$-\ln\mathbb{P}_Y(y) + \beta(1 - 2\mathbb{P}_Y(y) + \sum_{y'}\mathbb{P}_Y^2(y')) + \boldsymbol{\theta}^{\mathsf{T}}\boldsymbol{\phi}(\boldsymbol{x}, y) + Z_{\mathbb{Q}_Y}$$
$$+ \alpha(|\mathcal{Y}| - y + 1 + \sum_{i=1}^{|\mathcal{Y}|}[\sum_{j=1}^{i}\mathbb{P}_Y(j)]^2 - 2\sum_{i=1}^{|\mathcal{Y}|}\sum_{j=1}^{i}\mathbb{P}_Y(j) + 2\sum_{i=1}^{y-1}\sum_{j=1}^{i}\mathbb{P}_Y(j)) = 0. \tag{12}$$

Notice that $\sum_{i=1}^{|\mathcal{Y}|}[\sum_{j=1}^{i}\mathbb{P}_Y(j)]^2 - 2\sum_{i=1}^{|\mathcal{Y}|}\sum_{j=1}^{i}\mathbb{P}_Y(j)$ is constant across all $y$. By absorbing them into constant $C$, we also observe that the equation for $y$ only depends on $\mathbb{P}_Y(y')$ for $y' < y$ in the term $\sum_{i=1}^{y-1}\sum_{j=1}^{i}\mathbb{P}_Y(j)$. Therefore, $\mathbb{P}_Y^*(y)$ can be found in increasing order of $y$ from 1 to $|\mathcal{Y}|$.

Given $\mathbb{P}_Y^*$, consider Eq. (11) in matrix form:

$$\begin{pmatrix} -1/\mathbb{P}_Y(1) - 2\beta & 2\alpha & 4\alpha & \ldots & 2(|\mathcal{Y}| - 1)\alpha & 1 \\ 0 & -1/\mathbb{P}_Y(2) - 2\beta & 2\alpha & \ldots & 2(|\mathcal{Y}| - 2)\alpha & 1 \\ \ldots & \ldots & \ldots & \ldots & \ldots & \ldots \\ 0 & 0 & 0 & \ldots & -1/\mathbb{P}_Y(|\mathcal{Y}|) - 2\beta & 1 \\ 1 & 1 & 1 & \ldots & 1 & 0 \end{pmatrix}\begin{pmatrix} \mathbb{Q}_Y(1) \\ \mathbb{Q}_Y(2) \\ \ldots \\ \mathbb{Q}_Y(|\mathcal{Y}|) \\ Z_{\mathbb{P}_Y} \end{pmatrix} = \begin{pmatrix} C_1 \\ C_2 \\ \ldots \\ C_{|\mathcal{Y}|} \\ 1. \end{pmatrix}$$

This is not an unreduced Hessenberg matrix. However, notice that as $Z_{\mathbb{P}_Y}$ increases, $\mathbb{Q}_Y(|\mathcal{Y}|)$ also increases by the penultimate equation. This in turn increases $\mathbb{Q}_Y(|\mathcal{Y}| - 1)$ according to the third from last equation. Therefore, the solution $\mathbb{Q}_Y^*$ without the simplex constraint increases monotonically as $Z_{\mathbb{P}_Y}$ increases. We can use bisection method again to find the $\mathbb{Q}_Y^*$ that also satisfies the simplex constraint.

