# OpenReview forum: "Moment Distributionally Robust Probabilistic Supervised Learning"
_ICLR.cc/2023/Conference — Submitted to ICLR 2023_

### Official Review · Reviewer_ARxu · 2022-10-24

**Confidence:** 3
**Correctness:** 2
**Technical Novelty And Significance:** 1
**Empirical Novelty And Significance:** 1
**Recommendation:** 3

**Clarity, Quality, Novelty And Reproducibility:**

- Unclear mathematical exposition
- Limited novelty when compared with Li et al (2022)

**Strength And Weaknesses:**

The paper lacks clarity and mathematical rigor:
1. Before Proposition 1 in the main text, the authors claim that “the ambiguity set ($\mathcal{A}(\mathbb{P}^{\mathrm{emp}}$) is a compact convex set.” However, no proof is provided.
2. In the proof of Proposition 1, page 13, line 6, the authors invoke Sion’s minimax theorem. However, the authors did not verify that the loss function is lower-semicontinuous in $\mathbb{P}_{Y|X}$ and upper-semicontinuous in $\mathbb{Q}$.
3. Also, in the proof of Proposition 1, in the middle of page 14, the authors claimed that “Again by strong duality, we can rearrange…”. However, no justification for strong duality has been provided.
Further, I feel that the use of $I_{\mathcal{C}}(\cdot)$ is unnecessary. In fact, the indicator function is non-smooth, non-convex, which complicates the whole process of applying duality results.
4. The notations of $\min_{\mathbb{P}}$ is really confusing in the statement of Proposition 1. The expectation is already taken with respect to $P_x^{emp}$, so what is the $\min_{\mathbb{P}}$ in equation (3) concerning? Shouldn't $\mathbb{P}$ be $\mathbb{P}_{Y|X}$?
5. The results of this paper seem to follow from Li et al. (2022) [Proposition 1, Corollary 2 and Theorem 3], however, up to today (Oct 23), I could not find the manuscript of Li et al. (2022) on arXiv or on any public domain (openreview). At this point, it is difficult for me to judge the novelty of the manuscript.
6. Using the setup of this paper, any distribution can be conveniently represented as a matrix because the space $\mathcal{Y}$ is discrete and the space $\mathcal{X}$ can be reduced to a finite set supported on the training data. From this perspective, the problem becomes uninteresting because, in the end, we are only perturbing the elements of a matrix. The problem only becomes interesting, in my opinion, when $\mathcal{Y}$ is continuous or we allow perturbations of the marginal distribution on $X$.
7. The numerical experiments can not demonstrate that the proposed method is superior than existing approaches. The experiments are also conducted with $\varepsilon = 0$, which implies no moment robustness.
8. The authors should make clear how a probabilistic prediction is made at test time, when only $x$ is given and there is no information on $y$.

**Summary Of The Paper:**

This paper is concerned with probabilistic predictions: Instead of predicting a class, the prediction is a distribution over the label space. The paper proceeds by using a probabilistic loss functional, and consider an adversarial training approach. The proposed method is a minimax problem, with perturbations in a moment set that is specified using the feature function. The paper then provides the dual formulation and integrates this formulation into a gradient descent algorithm (Algorithm 1).

**Summary Of The Review:**

- The paper lacks mathematical rigor, which severely hinders my understanding and appreciation of the results.
- There is a significant overlap with Li et al (2022) which requires clarification.
- The experiment does not demonstrate the benefit of the proposed method

---

> ### Author Response · Authors · 2022-11-15
> **Response to Reviewer ARxu**
>
> Thanks for your comments.
>
> **Q1: Before Proposition 1 in the main text, the authors claim that “the ambiguity set ($\mathcal{A}(\mathbb{P}^{\text{emp}})$) is a compact convex set.” However, no proof is provided.**
>
> It is straightforward that the ambiguity set is the intersection of the simplex set and an affine subspace, which is a convex compact set.
>
> **Q2: In the proof of Proposition 1, page 13, line 6, the authors invoke Sion’s minimax theorem. However, the authors did not verify that the loss function is lower-semicontinuous in $\mathbb{P}_{Y|X}$  and upper-semicontinuous in \mathbb{Q}.**
>
> In line 11, page 4, we state our assumption that we are given a **continuous** proper loss.
>
> **Q3: Also, in the proof of Proposition 1, in the middle of page 14, the authors claimed that “Again by strong duality, we can rearrange…”. However, no justification for strong duality has been provided. Further, I feel that the use of $I_{\mathcal{C}}(\cdot)$ is unnecessary. In fact, the indicator function is non-smooth, non-convex, which complicates the whole process of applying duality results.**
>
> We assume that you mean page 13. The strong duality is obvious because it basically applies the same argument as in the first time we apply strong duality in lines 3-5, page 13. The use of an indicator function is a very standard technique to study constrained optimization problems when the domain of a function is restricted to a subset. The supremum over $\boldsymbol{u}$ is in the innermost level so all we do is push it to the outermost level without having to resort to strong duality.
>
> **Q4: The notations of $\min_{\mathbb{P}}$ is really confusing in the statement of Proposition 1. The expectation is already taken with respect to $\mathbb{P}\_{\boldsymbol{X}}^{\text{emp}}$, so what is the  in equation (3) concerning? Shouldn't $\mathbb{P}$ be $\mathbb{P}_{Y|\boldsymbol{X}}$?**
>
> You are correct on this point. We adopted $\mathbb{P}\_{Y|\boldsymbol{X}}$ in our original manuscript but revised it to simply $\mathbb{P}$ to avoid clutter of notation. $\mathbb{P}$ and $\mathbb{P}^{\text{emp}}$ are apparently different distributions and $\mathbb{P}$ denotes to the set of conditional distributions $\mathbb{P}\_{Y|\boldsymbol{X}}$ should be easily determined from context.
>
> **Q5: The results of this paper seem to follow from Li et al. (2022) [Proposition 1, Corollary 2 and Theorem 3], however, up to today (Oct 23), I could not find the manuscript of Li et al. (2022) on arXiv or on any public domain (openreview). At this point, it is difficult for me to judge the novelty of the manuscript.**
>
> The results in Li et al., 2022 are considered parallel to this work because it was made public (we may not be allowed to provide the link here and it should be found easily online by searching the title) last month. By a typical rule of thumb in computer science conference, works published 3 months before the submission deadline are considered contemporaneous. We cite this work for respecting parallel works but that does not indicate that the results in this paper are simple corollaries of the existing works. Furthermore, the problem setting is quite different (e.g., smooth losses vs structured losses). Specifically, Li et al., 2022 study a structured prediction problem with particular focus on tree-shaped objects with little discussion on probabilistic prediction and assuming point groundtruth. While we study probabilistic supervised learning and put our work in the literature of the more fundamental probability estimation and loss properness setting.
>
> **Q6: Using the setup of this paper, any distribution can be conveniently represented as a matrix because the space $\mathcal{Y}$  is discrete and the space $\mathcal{X}$ can be reduced to a finite set supported on the training data. From this perspective, the problem becomes uninteresting because, in the end, we are only perturbing the elements of a matrix. The problem only becomes interesting, in my opinion, when $\mathcal{Y}$ is continuous or we allow perturbations of the marginal distribution on $\mathcal{X}$.**
>
> This is a very brilliant point. The general probabilistic supervised learning refers to a mixture space $\mathcal{Y}$ of discrete and continuous sets. In the completely continuous case, the problem is actually called conditional density estimation, widely studied with non-parametric kernel methods. Perturbations on covariates are interesting as well but studying this might exceed the page limit of a conference paper submission. We will consider improve our manuscript based on this point.
>
> **Q7: The numerical experiments can not demonstrate that the proposed method is superior than existing approaches. The experiments are also conducted with $\varepsilon = 0$, which implies no moment robustness.**
>
> We agree that experimental results are not convincing. Although we set $\varepsilon = 0$ in the experiments, typical optimizers for neural networks usually impose implicit regularization.
>
> tbc

---

> > ### Author Response · Authors · 2022-11-15
> > **Response to Reviewer ARxu**
> >
> > **Q8: The authors should make clear how a probabilistic prediction is made at test time, when only $\boldsymbol{x}$ is given and there is no information on $y$.**
> >
> > The inference formula is given in the equation in the last statement of Corollary 2.
> >
> > **Additional comments**
> >
> > We hope that the above clarifications address most of your concerns. If your only concern is about the empirical results, it would be nice if you can reconsider the judgement on this paper. Given all the helpful reviews above, the decision on the manuscript is not that important to us. But please let us know if you have additional confusion with some other technical details, we are more than glad to have further discussion with you.

---

### Official Review · Reviewer_QSPn · 2022-10-25

**Confidence:** 3
**Correctness:** 4
**Technical Novelty And Significance:** 2
**Empirical Novelty And Significance:** 1
**Recommendation:** 5

**Clarity, Quality, Novelty And Reproducibility:**

This paper is well-organized and easy to follow. I found that some references might be related to this work but seems missing in the paper:

Shafieezadeh Abadeh, S., Mohajerin Esfahani, P. M., & Kuhn, D. (2015). Distributionally robust logistic regression. Advances in Neural Information Processing Systems, 28.
Lee, C., & Mehrotra, S. (2015). A distributionally-robust approach for finding support vector machines. Available from Optimization Online.
Chen, R., Hao, B., & Paschalidis, I. (2021). Distributionally Robust Multiclass Classification and Applications in Deep CNN Image Classifiers. arXiv preprint arXiv:2109.12772.
Amos, B., & Kolter, J. Z. (2017, July). Optnet: Differentiable optimization as a layer in neural networks. In International Conference on Machine Learning (pp. 136-145). PMLR.
Agrawal, A., Amos, B., Barratt, S., Boyd, S., Diamond, S., & Kolter, J. Z. (2019). Differentiable convex optimization layers. Advances in neural information processing systems, 32.



**Strength And Weaknesses:**

strength:
- A DRO version of probabilistic supervised learning is proposed under first-order moment ambiguity sets.
- The DRO problem can be incorporated into neural networks and perform end-to-end differentiable learning


Weaknesses
- The theoretical result is weak, and the technical contribution is not clear -- in all theorems and Propositions, the reference (Li et al., 2022) is cited, but there is no clear justification for the differences with (Li et al., 2022)
- The empirical study on real-world data is not enough, especially the baseline methods compared are very limited, only neural network models with the softmax and the spherical softmax function.


**Summary Of The Paper:**

This paper considers distributionally robust probabilistic supervised learning. The ambiguity set is constructed to include distributions that share the same marginal with the empirical distribution and are no more than ε away from the empirical in terms of first-order feature moment divergence. The strong duality is shown such that the primal DRO problem is equivalent to a regularized ERM problem. The authors characterize the solutions to the proposed method and present an efficient algorithm for specific losses. Moreover, neural network representations are incorporated and the DRO problem can serve as a differential layer to enable end-to-end differentiable learning.



**Summary Of The Review:**

Overall this paper is well-written and considers a problem that can be of broad interest to the machine-learning community. The results are standard -- such as the duality with regularized ERM and the risk bounds. However, the theoretical results look weak and most importantly, the authors didn't explain their technical contribution clearly, as compared with the work (Li et al., 2022). Moreover, the result in Corollary 2 is only limited to the case epsilon=0, which is the standard ERM, not the DRO problem.
Also considering that the numerical results are also limited and do not compare with many baseline methods, I think this is more like a borderline paper.

---

> ### Author Response · Authors · 2022-11-15
> **Response to Reviewer QSPn**
>
> Thanks for your helpful comments.
>
> **Q1: The theoretical result is weak, and the technical contribution is not clear -- in all theorems and Propositions, the reference (Li et al., 2022) is cited, but there is no clear justification for the differences with (Li et al., 2022)**
>
> The results in Li et al., 2022 are considered parallel to this work because it was made public last month. By a typical rule of thumb in computer science conference, works published 3 months before the submission deadline are considered contemporaneous. We cite this work for respecting parallel works but that does not indicate that the results in this paper are simple corollaries of the existing works. Furthermore, the problem setting is quite different (e.g., smooth losses vs structured losses). Specifically, Li et al., 2022 study a structured prediction problem with particular focus on tree-shaped objects with little discussion on probabilistic prediction and assuming point groundtruth. While we study probabilistic supervised learning and put our work in the literature of the more fundamental probability estimation and loss properness setting.
>
> **Q2: The empirical study on real-world data is not enough, especially the baseline methods compared are very limited, only neural network models with the softmax and the spherical softmax function.**
>
> We admit that the empirical results are not really convincing in its current form.
>
> **Q3: This paper is well-organized and easy to follow. I found that some references might be related to this work but seems missing in the paper:**
>
> We appreciate the reviewer for listing a few related works. We actually have studied most of them before proposing our method. We list the crucial differences with our work as follows:
>
> - Shafieezadeh Abadeh, S., Mohajerin Esfahani, P. M., & Kuhn, D. (2015). Distributionally robust logistic regression. Advances in Neural Information Processing Systems, 28. This is a seminal work that applies Wasserstein distributionally robust optimization to log loss minimization whereas we use moment DRO for the more general proper losses.
> - Lee, C., & Mehrotra, S. (2015). A distributionally-robust approach for finding support vector machines. Available from Optimization Online. This work uses Wasserstein distances as well. However, their method does not enable probabilistic classification and margin-based methods are well-known to be generally inconsistent.
> - Chen, R., Hao, B., & Paschalidis, I. (2021). Distributionally Robust Multiclass Classification and Applications in Deep CNN Image Classifiers. arXiv preprint arXiv:2109.12772. This work basically uses the same methodology in distributionally robust logistic regression and only optimizes the last linear layer in a NN model.
> - Amos, B., & Kolter, J. Z. (2017, July). Optnet: Differentiable optimization as a layer in neural networks. In International Conference on Machine Learning (pp. 136-145). PMLR. This work does not rely on DRO and simply minimizes the empirical risk.
> - Agrawal, A., Amos, B., Barratt, S., Boyd, S., Diamond, S., & Kolter, J. Z. (2019). Differentiable convex optimization layers. Advances in neural information processing systems, 32. Not DRO, same argument as above.
>
> **Q4: Moreover, the result in Corollary 2 is only limited to the case epsilon=0, which is the standard ERM, not the DRO problem.**
>
> The Fisher consistency results have to be achieved with $\varepsilon = 0$ because it obfuscates the original proper loss with a non-zero value. Note that this is a theoretical result with assumption on all measurable functions so it may not be called a limitation, because consistency results usually deal with an ideal setting with infinite training data, in which it is natural to impose no regularization.

---

### Official Review · Reviewer_u3yA · 2022-10-28

**Confidence:** 3
**Clarity, Quality, Novelty And Reproducibility:** See below.
**Correctness:** 4
**Technical Novelty And Significance:** 2
**Empirical Novelty And Significance:** 2
**Recommendation:** 5

**Strength And Weaknesses:**

See below.

**Summary Of The Paper:**

The authors present a method for learning a model that outputs probabilities wherein the output probability is the output of an optimization over an uncertainty set rather than, say the result of a softmax. They show how to learn a model using this approach and compare the method to simple alternatives on several datasets.

**Summary Of The Review:**

Overall, I found the technical aspects of this paper clear and well-written, and the methods and evaluations reasonable. With that said, I am still very unclear on the motivation for this method, especially given that if failed to outperform simple softmax on any of the evaluation problems. My concerns are below:

Major:

1. I did not follow the motivation for this method and strongly recommend that authors work to make the intuition clearer in the introduction.    Further, the experiments did nothing to make this motivation clearer. If anything, I left the paper wondering why I would use this method when a simple softmax appears to perform nearly identically. I recommend design their experiments to highlight and provide intuition for the differences between existing approaches and the proposed method and, if no differences exist, explain why a negative result is interesting. If there are no differences and no value in a negative result then it is hard for me to see the significance of the method.

Minor:

2. Define "proper" in the intro.

3. Contributions: "Extensive" seems strong here. Instead of saying "we performed evaluations", describe what your evaluations show and how that contributes to our knowledge.

---

> ### Author Response · Authors · 2022-11-15
> **Response to Reviewer u3yA**
>
> Thanks for your very insightful comments. We agree with your concerns by now and we will definitely incorporate your suggestions in our revision.

---

### Official Review · Reviewer_FSDY · 2022-10-30

**Confidence:** 2
**Correctness:** 3
**Technical Novelty And Significance:** 3
**Empirical Novelty And Significance:** 3
**Recommendation:** 3

**Clarity, Quality, Novelty And Reproducibility:**

The paper is cleanly written by the novelty seems to be the algorithm to compute the DRO alone, which I believe is a useful thing.

**Strength And Weaknesses:**

The derivation in the paper is theoretically interesting.

My main concern is this:

While the paper guarantees $\theta$ is learned well with $\phi$ fixed, it is unclear to me why I can learn a representation with the given objective. The reason is that changing $\phi$ changes the uncertainty set for the setup DRO problem! Without the ability to use a fixed uncertainty set from some known restriction, what is the value of the proposed method? The experiments seem to support this concern as the authors say "All the methods become vulnerable for large pnoise possibly because of the backbone neural network model."


Weaknesses/Questions:
1. Both corrollary 2 and theorem 3.2, along with the stated uncertainty set are from ((Li et al., 2022). Why do the authors claim " We propose a distributionally robust probabilistic supervised learning method"? The main contribution seems to be the algorithm.
2. What can go wrong when let $\beta=0$?
3. Theoretically, it seems like when learning $\phi$, it seems possible to end up in a minima where $\phi=0$. Can the authors explain why this won't happen when the objective is optimized?
4. Is mixing log-likelihood and brier score used in an application? Can the authors point to a case?
5. The experiments in table 1 do not seem to favor the proposed method much; softmax is better or similar.

**Summary Of The Paper:**

The paper proposes an algorithm to compute a DRO objective where the uncertainty set is defined with deviations in moments of the representation. The algorithm allows one to compute the minimax DRO loss using a primal-dual formulation and reasoning about the stationary points.

**Summary Of The Review:**

While I think the ability to compute minimax loss is great, I am unclear about some details and what the value of the paper is.

---

> ### Author Response · Authors · 2022-11-15
> **Response to Reviewer FSDY**
>
> Thanks for your comments.
>
> **Q1: While the paper guarantees $\boldsymbol{\theta}$ is learned well with $\boldsymbol{\phi}$ fixed, it is unclear to me why I can learn a representation with the given objective.**
>
> Our methods build upon first principles in a DRO formulation based on ambiguity sets defined by feature moments with fixed feature functions. Our contributions do **not** focus on learning a representation up to section 3.3. In this way, it is guaranteed that we end up with a convex optimization problem with global convergence guarantees and Fisher consistency results. We consider the extension of its ability to enable end-to-end representation learning as a plus of this family of DRO methods. Now, consider the setting where we relax our assumption on a fixed feature mapping. The formulation can be rewritten as
> $\min_{\boldsymbol{\theta}} \max_{\mathbb{Q} \in \mathcal{A}_{\boldsymbol{\theta}}(\mathbb{P}^{\text{emp}})} L^{\text{adv}}(\boldsymbol{\theta}, \mathbb{Q})$,
> which is a DRO problem with a **decision-dependent** ambiguity set, still an active research topic [1, 2, 3, 4, 5]. The uncertainty inherent in the ambiguity set itself is called endogenous uncertainty. A similar work adopting a similar ambiguity set to our method optimizes the multi-variate performance metric with this kind of representation learning [6].
>
> **Q2: Both corrollary 2 and theorem 3.2, along with the stated uncertainty set are from ((Li et al., 2022). Why do the authors claim " We propose a distributionally robust probabilistic supervised learning method"? The main contribution seems to be the algorithm.**
>
> The results in Li et al., 2022 are considered parallel to this work because it was made public last month. By a typical rule of thumb in computer science conference, works published 3 months before the submission deadline are considered contemporaneous. We cite this work for respecting parallel works but that does not indicate that the results in this paper are simple corollaries of the existing works. Furthermore, the problem setting is quite different (e.g., smooth losses vs structured losses).
>
> **Q3: What can go wrong when let ?**
>
> Nothing will go wrong. It simply reduces to robust logistic regression.
>
> **Q4: Theoretically, it seems like when learning $\boldsymbol{\phi}$, it seems possible to end up in a minima where $\boldsymbol{\phi} = 0$. Can the authors explain why this won't happen when the objective is optimized?**
>
> Without the ambiguity set, or equivalent set the radius to zero, our method reduces to empirical risk minimization with a proper loss. In practice, even non-linear hypothesis such as neural networks are restrictive. The minima of a proper loss is well-known [7], which also serves a lower bound for our adversarial loss. That's why the seemingly trivial solution $\boldsymbol{\phi} = 0$ you mentioned is often not optimal.
>
> **Q5: Is mixing log-likelihood and brier score used in an application? Can the authors point to a case?**
>
> First, we mix them because we still would like a proper loss for correct probability estimation. Second, an application could be estimating an event that has severe consequences such as surgeon planning, disaster or earthquake prediction, in which overestimating the probability of an unexpected event happening is favored to underestimation.
>
> **Q6: The experiments in table 1 do not seem to favor the proposed method much; softmax is better or similar.**
>
> That's true. But we hope that if we address your concerns above, it would be nice if your judgement is not solely based on the experimental results.
>
> [1] Luo, Fengqiao, and Sanjay Mehrotra. "Distributionally robust optimization with decision dependent ambiguity sets." Optimization Letters 14, no. 8 (2020): 2565-2594.
>
> [2] Noyan, Nilay, Gábor Rudolf, and Miguel Lejeune. "Distributionally Robust Optimization Under a Decision-Dependent Ambiguity Set with Applications to Machine Scheduling and Humanitarian Logistics." INFORMS Journal on Computing 34, no. 2 (2022): 729-751.
>
> [3] Royset, Johannes O., and Roger J-B. Wets. "Variational theory for optimization under stochastic ambiguity." SIAM Journal on Optimization 27, no. 2 (2017): 1118-1149.
>
> [4] Zhang, Jie, Huifu Xu, and Liwei Zhang. "Quantitative stability analysis for distributionally robust optimization with moment constraints." SIAM Journal on Optimization 26, no. 3 (2016): 1855-1882.
>
> [5] Doan, Xuan Vinh. "Distributionally robust optimization under endogenous uncertainty with an application in retrofitting planning." European Journal of Operational Research 300, no. 1 (2022): 73-84.
>
> [6] Fathony, Rizal, and Zico Kolter. "AP-perf: Incorporating generic performance metrics in differentiable learning." In International Conference on Artificial Intelligence and Statistics, pp. 4130-4140. PMLR, 2020.
>
> [7] Williamson, Robert, Elodie Vernet, and Mark Reid. "Composite multiclass losses." (2016).

---

### Decision · Program_Chairs · 2023-01-20

**Decision:**

Reject

**Justification For Why Not Higher Score:**

The paper did not receive support from any of the reviewers. The method is not properly motivated and shows no empirical advantage over simple baselines.

**Justification For Why Not Lower Score:**

N/A

**Metareview: Summary, Strengths And Weaknesses:**

The paper proposes a method for predicting distributions in the DRO setting. The reviewers had several concerns, with a particular focus on lack of clear motivation, and empirical results which do not demonstrate an advantage for the method.